# Explicit ion modeling predicts physicochemical interactions for chromatin organization

Xingcheng Lin[†, ‡], Bin Zhang*

Department of Chemistry, Massachusetts Institute of Technology, Cambridge, United States

**\*For correspondence:**
binz@mit.edu

**Present address:** [†]Department of Physics, North Carolina State University, Raleigh, United States; [‡]Bioinformatics Research Center, North Carolina State University, Raleigh, United States

**Abstract** Molecular mechanisms that dictate chromatin organization in vivo are under active investigation, and the extent to which intrinsic interactions contribute to this process remains debatable. A central quantity for evaluating their contribution is the strength of nucleosome-nucleosome binding, which previous experiments have estimated to range from 2 to 14 $k_BT$. We introduce an explicit ion model to dramatically enhance the accuracy of residue-level coarse-grained modeling approaches across a wide range of ionic concentrations. This model allows for de novo predictions of chromatin organization and remains computationally efficient, enabling large-scale conformational sampling for free energy calculations. It reproduces the energetics of protein-DNA binding and unwinding of single nucleosomal DNA, and resolves the differential impact of mono- and divalent ions on chromatin conformations. Moreover, we showed that the model can reconcile various experiments on quantifying nucleosomal interactions, providing an explanation for the large discrepancy between existing estimations. We predict the interaction strength at physiological conditions to be 9 $k_BT$, a value that is nonetheless sensitive to DNA linker length and the presence of linker histones. Our study strongly supports the contribution of physicochemical interactions to the phase behavior of chromatin aggregates and chromatin organization inside the nucleus.

## eLife assessment

The authors have developed a **compelling** coarse-grained simulation approach for nucleosome-nucleosome interactions within a chromatin array. The data presented are **solid** and provide new insights that allow for predictions of how chromatin interactions might occur in vivo. The tools presented herein will be **valuable** for the chromosome biology field.

## Introduction

Three-dimensional genome organization plays essential roles in numerous DNA-templated processes (*Dekker et al., 2013*; *Bonev and Cavalli, 2016*; *Finn and Misteli, 2019*; *Misteli, 2020*; *Lin et al., 2021b*). Understanding the molecular mechanisms for its establishment could improve our understanding of these processes and facilitate genome engineering. Advancements in high-throughput sequencing and microscopic imaging have enabled genome-wide structural characterization, revealing a striking compartmentalization of chromatin at large scales (*Lieberman-Aiden et al., 2009*; *Quinodoz et al., 2018*; *Su et al., 2020*; *Takei et al., 2021*). For example, A compartments are enriched with euchromatin and activating post-translational modifications to histone proteins. They are often spatially segregated from B compartments that enclose heterochromatin with silencing histone marks (*Gibcus and Dekker, 2013*; *Finn and Misteli, 2019*; *Misteli, 2020*; *Mirny and Dekker, 2022*; *Xie and Zhang, 2019*).

Compartmentalization has been proposed to arise from the microphase separation of different chromatin types as in block copolymer systems (*Fujishiro and Sasai, 2022*; *Jost et al., 2014*; *Falk et al., 2019*; *Bajpai et al., 2021*; *Laghmach et al., 2020*; *Hu et al., 2013*; *Lesne et al., 2014*; *Di Pierro et al., 2016*; *Xie et al., 2017*; *Yildirim and Feig, 2018*; *MacPherson et al., 2018*; *Shi and Thirumalai, 2021*; *Brahmachari et al., 2022*). However, the molecular mechanisms that drive the microphase separation are not yet fully understood. Protein molecules that recognize specific histone modifications have frequently been found to undergo liquid-liquid phase separation (*Larson et al., 2017*; *Kent et al., 2020*; *Xie et al., 2022*; *Leicher et al., 2022*; *Latham and Zhang, 2021*; *Lin et al., 2021a*; *MacPherson et al., 2018*), potentially contributing to chromatin demixing. Demixing can also arise from interactions between chromatin and various nuclear landmarks such as nuclear lamina and speckles (*Brahmachari et al., 2022*; *Falk et al., 2019*; *Mirny and Dekker, 2022*; *Kamat et al., 2023*), as well as active transcriptional processes (*Hilbert et al., 2021*; *Jiang et al., 2022*; *Brahmachari et al., 2023*; *Goychuk et al., 2023*). Furthermore, recent studies have revealed that nucleosome arrays alone can undergo spontaneous phase separation (*Gibson et al., 2019*; *Strickfaden et al., 2020*; *Zhang et al., 2022*), indicating that compartmentalization may be an intrinsic property of chromatin driven by nucleosome-nucleosome interactions.

The relevance of physicochemical interactions between nucleosomes to chromatin organization in vivo has been constantly debated, partly due to the uncertainty in their strength (*Kruithof et al., 2009*; *Cui and Bustamante, 2000*; *Kaczmarczyk et al., 2020*; *Funke et al., 2016*). Examining the interactions between native nucleosomes poses challenges due to the intricate chemical modifications that histone proteins undergo within the nucleus and the variations in their underlying DNA sequences (*Fenley et al., 2010*; *Fenley et al., 2018*). Many in vitro experiments have opted for reconstituted nucleosomes that lack histone modifications and feature well-positioned 601-sequence DNA (*Lowary and Widom, 1998*) to simplify the chemical complexity. These experiments aim to establish a fundamental reference point, a baseline for understanding the strength of interactions within native nucleosomes. Nevertheless, even with reconstituted nucleosomes, a consensus regarding the significance of their interactions remains elusive. For example, using force-measuring magnetic tweezers, Kruithof et al. estimated the inter-nucleosome binding energy to be ~14 $k_BT$ (*Kruithof et al., 2009*). On the other hand, Funke et al. introduced a DNA-origami-based force spectrometer to directly probe the interaction between a pair of nucleosomes (*Funke et al., 2016*), circumventing any potential complications from interpretations of single-molecule traces of nucleosome arrays. Their measurement reported a much weaker binding free energy of approximately 2 $k_BT$. This large discrepancy in the reported reference values complicates a further assessment of the interactions between native nucleosomes and their contribution to chromatin organization in vivo.

Computational modeling is well suited for reconciling the discrepancy across experiments and determining the strength of inter-nucleosome interactions. The high computational cost of atomistic simulations (*Winogradoff et al., 2015*; *Woods et al., 2021*; *Li et al., 2023*) has inspired several groups to calculate the nucleosome binding free energy with coarse-grained models (*Moller et al., 2019*; *Farr et al., 2021*). However, the complex distribution of charged amino acids and nucleotides at nucleosome interfaces places a high demand on force field accuracy. In particular, most existing models adopt a mean-field approximation with the Debye-Hückel theory (*Phillips, 2012*) to describe electrostatic interactions in an implicit-solvent environment (*Izadi et al., 2016*; *Bascom and Schlick, 2018*; *Moller et al., 2019*; *Farr et al., 2021*), preventing an accurate treatment of the complex salt conditions explored in experiments. Further force field development is needed to improve the accuracy of coarse-grained modeling across different experimental settings (*Freeman et al., 2011*; *Hinckley and de Pablo, 2015*; *Sun et al., 2022*; *Hayes et al., 2015*).

We introduce a residue-level coarse-grained explicit ion model for simulating chromatin conformations and quantifying inter-nucleosome interactions. We validate our model's accuracy through extensive simulations, demonstrating that it reproduces the binding affinities of protein-DNA complexes (*Privalov et al., 2011*) and energetic cost of nucleosomal DNA unwinding (*Hall et al., 2009*). Further simulations of chromatin at various salt concentrations reproduce experimentally measured sedimentation coefficients (*Correll et al., 2012*). We also reveal extensive close contacts between histone proteins and DNA across nucleosomes, the perturbation of which explains the discrepancy among various experimental studies. Finally, we determined the binding free energy between a pair of nucleosomes under physiological salt concentrations as ~9 $k_BT$. While longer linker DNA would reduce this

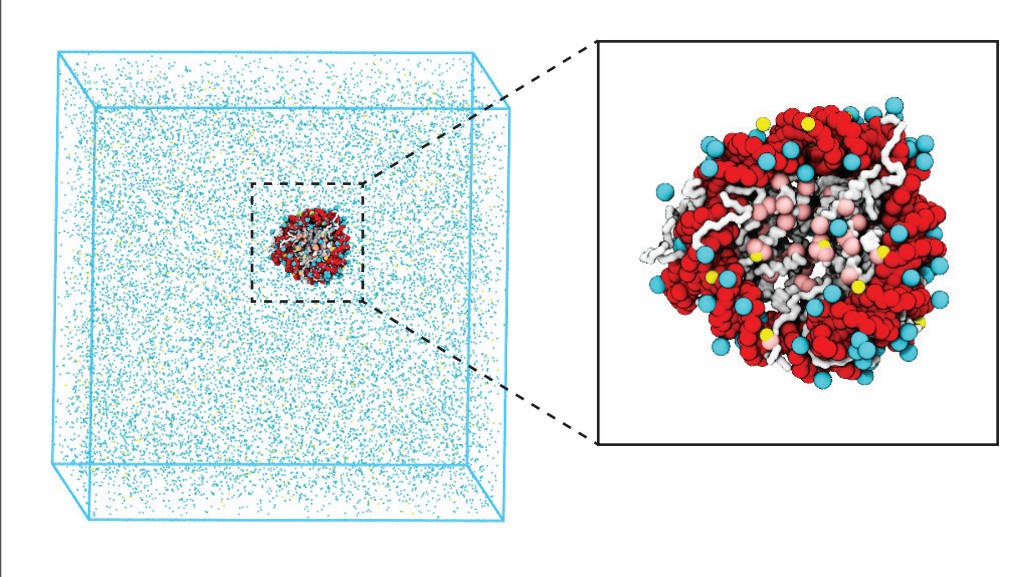

**Figure 1.** Illustration of the residue-level coarse-grained explicit ion model for chromatin simulations. The left panel presents a snapshot for the simulation box of a 147 bp nucleosome in a solution of 100 mM NaCl and 0.5 mM $MgCl_2$. The nucleosomal DNA and histone proteins are colored in red and white, respectively. The zoom-in on the right highlights the condensation of ions around the nucleosome, with $Na^+$ in cyan and $Mg^{2+}$ in yellow. Negative residues of the histone proteins are colored in pink.

binding energy, linker histones can more than compensate this reduction to mediate inter-nucleosome interactions with disordered, charged terminal tails. Our study supports the importance of intrinsic physicochemical interactions in chromatin organization in vivo.

## Results

### Counterion condensation accommodates nucleosomal DNA unwrapping

Various single-molecule studies have been carried out to probe the stability of nucleosomes and the interactions between histone proteins and DNA (*Bennink et al., 2001*; *Cui and Bustamante, 2000*; *Pope et al., 2005*; *Bancaud et al., 2007*; *Hall et al., 2009*). The DNA-unzipping experiment performed by *Hall et al., 2009*, is particularly relevant since the measured forces can be converted into a free energy profile of DNA unwinding at a base-pair resolution, as shown by Forties et al. with a continuous-time Markov model (*Forties et al., 2011*). The high-resolution quantification of nucleosome energetics is valuable for benchmarking the accuracy of computational models.

We introduce a coarse-grained explicit ion model for chromatin simulations (*Figure 1*). The model represents each amino acid with one coarse-grained bead and three beads per nucleotide. It resolves the differences among various chemical groups to accurately describe biomolecular interactions with physical chemistry potentials. Our explicit representation of monovalent and divalent ions enables a faithful description of counterion condensation and its impact on electrostatic interactions between protein and DNA molecules. Additional model details are provided in the Materials and methods and Appendix.

We performed umbrella simulations (*Torrie and Valleau, 1977*) to determine the free energy profile of nucleosomal DNA unwinding. The experimental buffer condition of 0.10 M NaCl and 0.5 mM $MgCl_2$ (*Hall et al., 2009*) was adopted in simulations for direct comparison. As shown in *Figure 2B*, the simulated values match well with experimental results over a wide range. Furthermore, we computed the binding free energy for a diverse set of protein-DNA complexes and the simulated values again match well with experimental data (*Figure 2—figure supplement 1*), supporting the model's accuracy.

Counterions are often released upon protein-DNA binding to make room for close contacts at the interface, contributing favorably to the binding free energy in the form of entropic gains (*Schiessel, 2003*). However, previous studies have shown that the histone-DNA interface in a fully wrapped

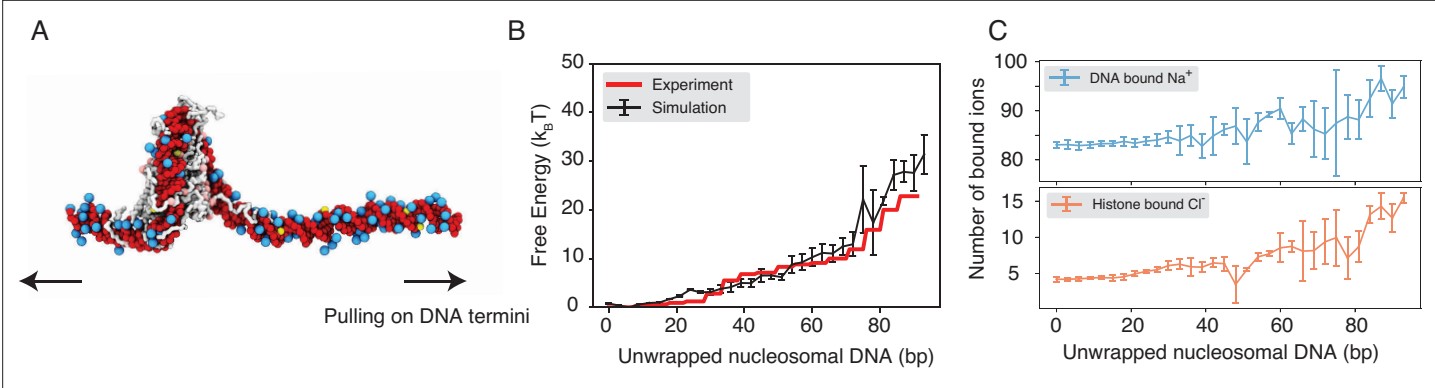

**Figure 2.** Explicit ion modeling reproduces the energetics of nucleosomal DNA unwrapping. (**A**) Illustration of the umbrella simulation setup using the end-to-end distance between two DNA termini as the collective variable. The same color scheme as in *Figure 1* is adopted. Only ions close to the nucleosomes are shown for clarity. (**B**) Comparison between simulated (black) and experimental (red) free energy profile as a function of the unwrapped DNA base pairs. Error bars were computed as the standard deviation of three independent estimates. (**C**) The average number of Na⁺ ions within 10 Å of the nucleosomal DNA (top) and Cl⁻ions within 10 Å of histone proteins (bottom) are shown as a function of the unwrapped DNA base pairs. Error bars were computed as the standard deviation of three independent estimates.

The online version of this article includes the following figure supplement(s) for figure 2:

**Figure supplement 1.** The explicit ion model predicts the binding affinities of protein-DNA complexes well, related to *Figure 1* of the main text.

nucleosome configuration is not tightly sealed but instead permeated with water molecules and mobile ions (*Davey et al., 2002*; *Materese et al., 2009*). Given their presence in the bound form, how these counterions contribute to nucleosomal DNA unwrapping remains to be shown. We calculated the number of DNA-bound cations and protein-bound anions as DNA unwraps. Our results, shown in *Figure 2C*, indicate that only a modest amount of extra Na⁺ and Cl⁻ ions becomes associated with the nucleosome as the outer DNA layer unwraps. However, significantly more ions become bound when the inner layer starts to unwrap (after 73 bp). These findings suggest that counterion release may contribute more significantly to the inner layer wrapping, potentially caused by a tighter protein-DNA interface.

## Charge neutralization with Mg²⁺ compacts chromatin

In addition to contributing to the stability of individual nucleosomes, counterions can also impact higher-order chromatin organization. Numerous groups have characterized the structures of nucleosome arrays (*Widom, 1986*; *Schwarz et al., 1996*; *Engelhardt, 2004*; *Correll et al., 2012*; *Grigoryev et al., 2009*; *Allahverdi et al., 2015*), revealing a strong dependence of chromatin folding on the concentration and valence of cations.

To further understand the role of counterions in chromatin organization, we studied a 12-mer with 20-bp-long linker DNA under different salt conditions. We followed the experiment setup by *Correll et al., 2012*, that immerses chromatin in solutions with 5 mM NaCl, 150 mM NaCl, 0.6 mM MgCl₂, or 1 mM MgCl₂. To facilitate conformational sampling, we carried out umbrella simulations with a collective variable that quantifies the similarity between a given configuration and a reference two-start helical structure. Simulation details and the precise definition of the collective variable are provided in the Materials and methods and Appendix. Data from different umbrella windows were combined together with proper reweighting (*Kumar et al., 1992*) for analysis.

As shown in *Figure 3A*, the average sedimentation coefficients determined from our simulations match well with experimental values. Specifically, the simulations reproduce the strong contrast in chromatin size between the two systems with different NaCl concentrations. Chromatin under 5 mM NaCl features an extended configuration with minimal stacking between one and three nucleosomes (*Figure 3B*). On the other hand, the compaction is evident at 150 mM NaCl. Notably, in agreement with previous studies (*Ding et al., 2021*; *Liu et al., 2022*; *Cai et al., 2018*; *Dombrowski et al., 2022*), we observe tri-nucleosome configurations as chromatin extends. Finally, the simulations also support that divalent ions are more effective in packaging chromatin than NaCl. Even in the presence of

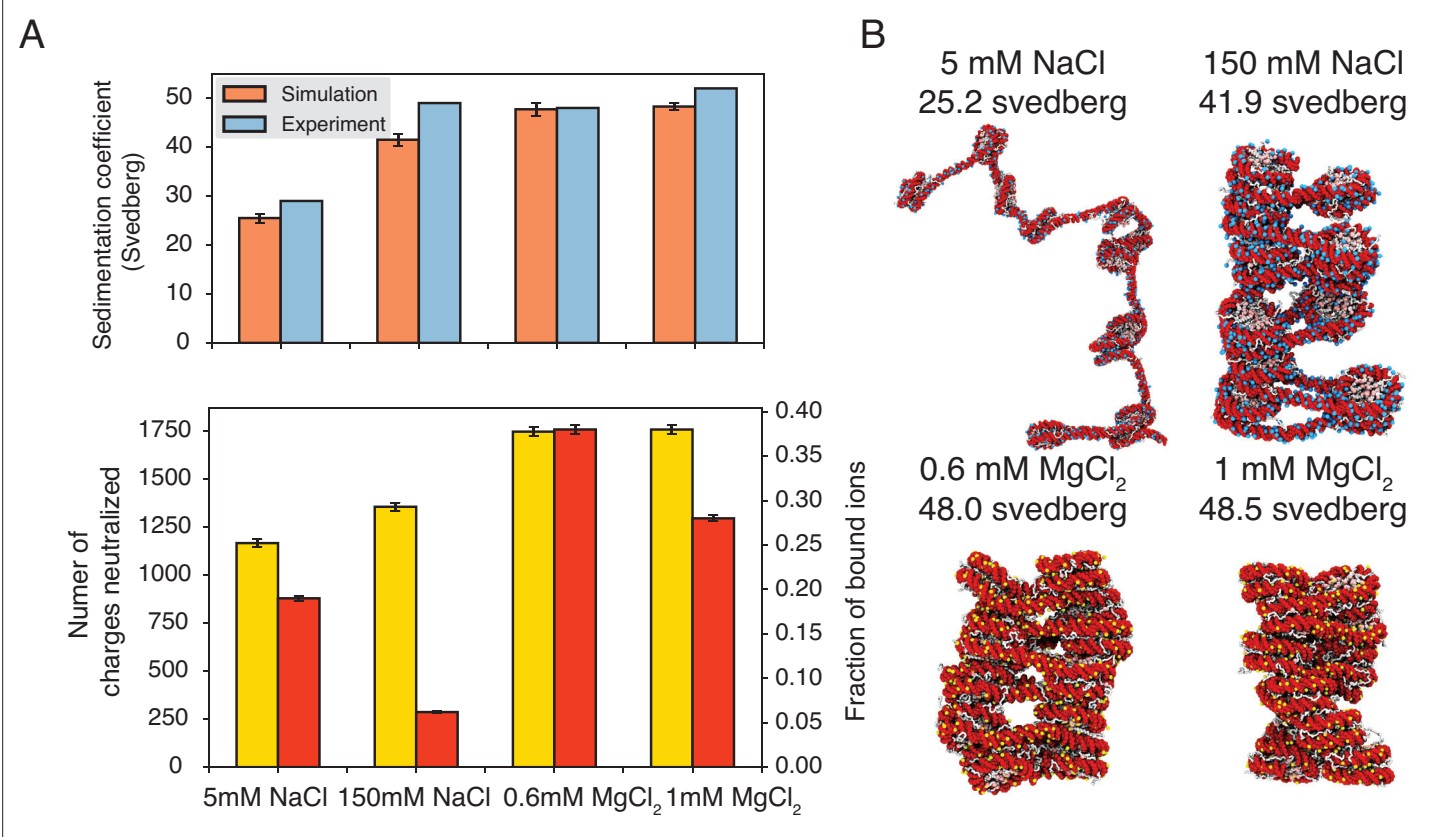

**Figure 3.** Explicit ion modeling predicts salt-dependent conformations of a 12-mer nucleosome array. (**A**) Top: Comparison of simulated and experimental (*Correll et al., 2012*) sedimentation coefficients of chromatin at different salt concentrations. Bottom: Number of DNA charges neutralized by bound cations (yellow, left y-axis label) and the fraction of ions bound to DNA (red, right y-axis label) at different salt concentrations. The error bars were estimated from the standard deviation of simulated probability distributions (*Figure 3—figure supplement 1*). (**B**) Representative chromatin structures with sedimentation coefficients around the mean values at different salt concentrations.

The online version of this article includes the following figure supplement(s) for figure 3:

**Figure supplement 1.** Probability distributions used to compute means and standard deviations of the quantities presented in *Figure 3* of the main text.

0.6 mM MgCl$_2$, the chromatin sedimentation coefficient is comparable to that obtained at 150 mM of NaCl.

We further characterized ions that are in close contact with DNA to understand their impact on chromatin organization. Our simulations support the condensation of cations, especially for divalent ions (*Figure 3A*, bottom) as predicted by the Manning theory (*Manning, 1978*; *Clark and Kimura, 1990*). Ion condensation weakens the repulsion among DNA segments that prevents chromatin from collapsing. Notably, the fraction of bound Mg$^{2+}$ is much higher than Na$^+$. Correspondingly, the amount of neutralized negative charges is always greater in systems with divalent ions, despite the significantly lower salt concentrations. The difference between the two types of ions arises from the more favorable interactions between Mg$^{2+}$ and phosphate groups that more effectively offset the entropy loss due to ion condensation (*Clark and Kimura, 1990*). While higher concentrations of NaCl do not dramatically neutralize more charges, the excess ions provide additional screening to weaken the repulsion among DNA segments, stabilizing chromatin compaction.

## Close contacts drive nucleosome binding free energy

Encouraged by the explicit ion model's accuracy in reproducing experimental measurements of single-nucleosomes and nucleosome arrays, we moved to directly quantify the strength of inter-nucleosomes interactions. We once again focus on reconstituted nucleosomes for a direct comparison with in vitro experiments. These experiments have yielded a wide range of values, ranging from 2 to 14 $k_BT$ (*Funke*

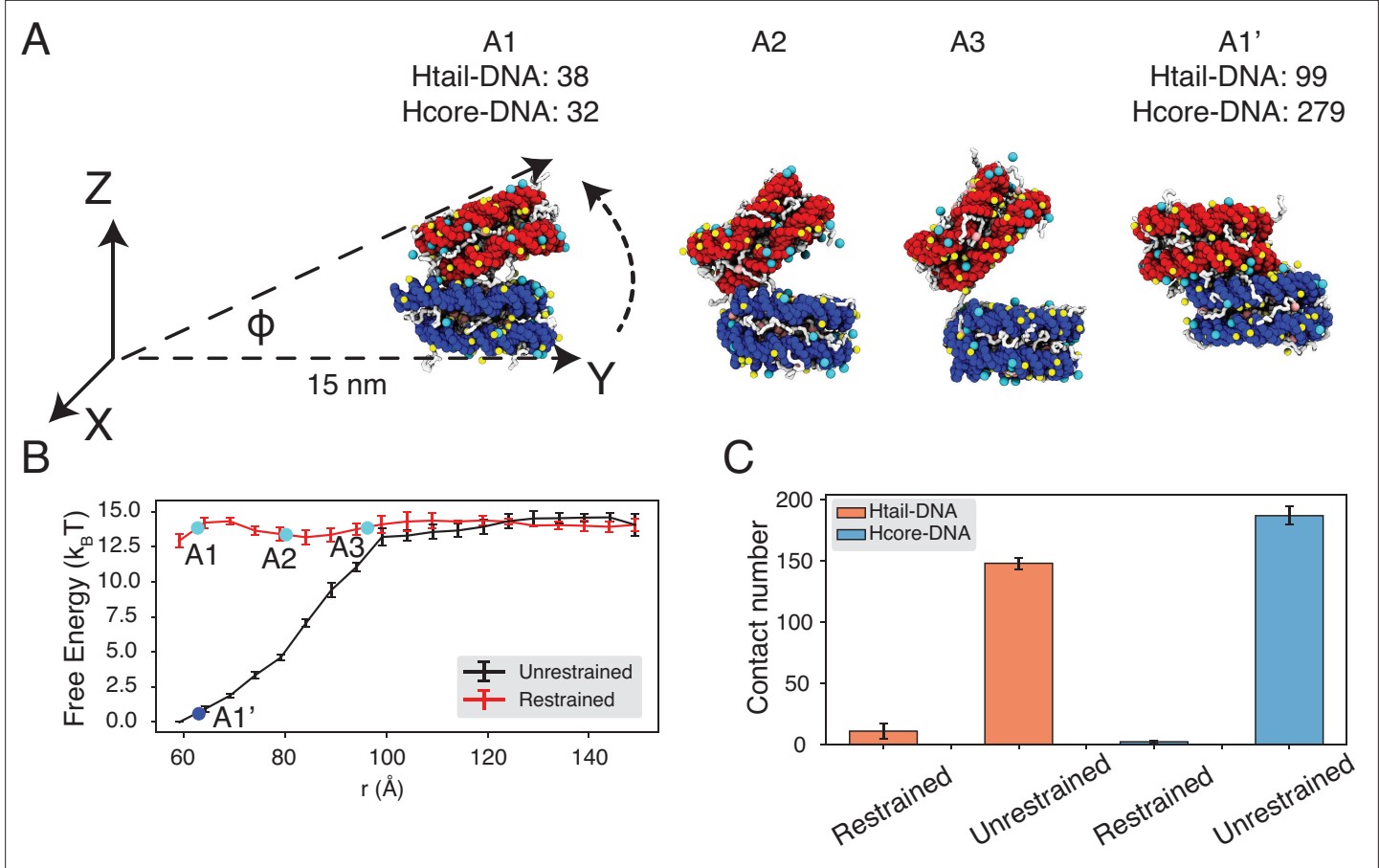

**Figure 4.** Close contacts give rise to strong inter-nucleosomal interactions. (**A**) Illustration of the simulation protocol employed to mimic the nucleosome unbinding pathway dictated by the DNA-origami device (**Funke et al., 2016**). The three configurations, A1, A2, and A3, corresponding to the three cyan dots in part B at distances 62.7, 80.2, and 96.3 Å. For comparison, a tightly bound configuration uncovered in simulations without any restraints of nucleosome movement is shown as A1'. The number of contacts formed by histone tails and DNA (Htail-DNA) and by histone core and DNA (Hcore-DNA) from different nucleosomes is shown for A1 and A1'. (**B**) Free energy profile as a function of the distance between the geometric centers of the two nucleosomes, computed from unrestrained (black) and DNA-origami-restrained simulations (red). Error bars were computed as the standard deviation of three independent estimates. (**C**) Average inter-nucleosomal contacts between DNA and histone tail (orange) and core (blue) residues, computed from unrestrained and DNA-origami-restrained simulations. Error bars were computed as the standard deviation of three independent estimates.

The online version of this article includes the following figure supplement(s) for figure 4:

**Figure supplement 1.** Illustration of the restrained two nucleosome simulations setup, related to **Figure 4** of the main text.

**Figure supplement 2.** Explicit ion modeling reproduces the experimental free energy profiles of nucleosome binding.

**Figure supplement 3.** Compared with DNA-origami-restrained simulations, the unrestrained simulations produce more histone-DNA contacts across nucleosomes, related to **Figure 4** of the main text.

**Figure supplement 4.** The unrestricted simulations favor a smaller angle $\theta$ between two nucleosomal planes compared to the DNA-origami-restrained simulations, related to **Figure 4** of the main text.

et al., 2016; **Cui and Bustamante, 2000**; **Kruithof et al., 2009**). Accurate quantification will offer a reference value for conceptualizing the significance of physicochemical interactions among native nucleosomes in chromatin organization in vivo.

To reconcile the discrepancy among various experimental estimations, we directly calculated the binding free energy between a pair of nucleosomes with umbrella simulations. We adopted the same ionic concentrations as in the experiment performed by **Funke et al., 2016**, with 35 mM NaCl and 11 mM $MgCl_2$. We focus on this study since the experiment directly measured the inter-nucleosomal interactions, allowing straightforward comparison with simulation results. Furthermore, the reported value for nucleosome binding free energy deviates the most from other studies. In one set of umbrella

simulations, we closely mimicked the DNA-origami device employed by Funke et al. to move nucleo-somes along a predefined path for disassociation (*Figure 4A*, A1 to A3). For example, neither nucle-osome can freely rotate (*Figure 4—figure supplement 1*); the first nucleosome is restricted to the initial position, and the second nucleosome can only move within the *Y-Z* plane along the arc 15 nm away from the origin. For comparison, we performed a second set of independent simulations without imposing any restrictions on nucleosome orientations. Additional simulation details can be found in Materials and methods and Appendix.

Strikingly, the two sets of simulations produced dramatically different binding free energies. Restricting nucleosome orientations produced a binding free energy of ~2 $k_B T$, reproducing the experimental value (*Figure 4B*, *Figure 4—figure supplement 2*). On the other hand, the binding free energy increased to 15 $k_B T$ upon removing the constraints.

Further examination of inter-nucleosomal contacts revealed the origin of the dramatic difference in nucleosome binding free energies. As shown in *Figure 4C*, the average number of contacts formed between histone tails and DNA from different nucleosomes is around 150 and 10 in the two sets of simulations. A similar trend is observed for histone core-DNA contacts across nucleosomes. The differences are most dramatic at small distances (*Figure 4B*, *Figure 4—figure supplement 3*) and are clearly visible in the most stable configurations. For example, from the unrestricted simulations, the most stable binding mode corresponds to a configuration in which the two nucleosomes are almost parallel to each other (see configuration A1' in *Figure 4A*), with the angle between the two nucleo-some planes close to zero (*Figure 4B*, *Figure 4—figure supplement 4*). However, the inherent design of the DNA-origami device renders this binding mode inaccessible, and the smallest angle between the two nucleosome planes is around 23° (see configuration A1 in *Figure 4A*). Therefore, a significant loss of inter-nucleosomal contacts caused the small binding free energy seen experimentally.

## Modulation of nucleosome binding free energy by in vivo factors

The predicted strength for unrestricted inter-nucleosome interactions supports their significant contri-bution to chromatin organization in vivo. However, the salt concentration studied above and in the DNA-origami experiment is much higher than the physiological value (*Kaczmarczyk et al., 2020*). To further evaluate the in vivo significance of inter-nucleosome interactions, we computed the binding free energy at the physiological salt concentration of 150 mM NaCl and 2 mM of $MgCl_2$.

We observe a strong dependence of nucleosome orientations on the inter-nucleosome distance. A collective variable, $\theta$, was introduced to quantify the angle between the two nucleosomal planes (*Figure 5A*). As shown in two-dimensional binding free energy landscape of inter-nucleosome distance, $r$, and $\theta$ (*Figure 5B*), at small distances (~60 Å), the two nucleosomes prefer a face-to-face binding mode with small $\theta$ values. As the distance increases, the nucleosomes will almost undergo a 90° rotation to adopt perpendicular positions. Such orientations allow the nucleosomes to remain in contact and is more energetically favorable. The orientation preference gradually diminishes at large distances once the two nucleosomes are completely detached. Importantly, we observed a strong inter-nucleosomal interaction with two nucleosomes wrapped by 147 bp 601-sequence DNA (~9 $k_B T$).

Furthermore, we found that the nucleosome binding free energy is minimally impacted by the precise DNA sequence. For example, when the 601 sequence is replaced with poly-dA:dT or poly-dG:dC, the free energy only varied by ~2 $k_B T$ (*Figure 5—figure supplement 1*). However, the poly-dA:dT sequence produced stronger binding while poly-dG:dC weakened the interactions. The sequence specific effects are potentially due to the increased stiffness of poly-dA:dT DNA (*Ortiz and de Pablo, 2011*), which causes the DNA to unwrap more frequently, increasing cross-nucleosome contacts at larger distances (*Figure 5—figure supplement 2*).

In addition to variations in DNA sequences, in vivo nucleosomes also feature different linker lengths. We performed simulations that extend the 601 sequence with 10 extra base pairs of poly-dA:dT sequence at each end, reaching a nucleosome repeat length (NRL) of 167 bp. Consistent with previous studies (*Mangenot et al., 2002*; *Correll et al., 2012*; *Huang et al., 2018*), increasing the NRL weakened inter-nucleosomal interactions (*Figure 5C* and *Figure 5—figure supplement 3*), reducing the binding free energy to ~6 $k_B T$.

Importantly, we found that the weakened interactions upon extending linker DNA can be more than compensated for by the presence of histone H1 proteins. This is demonstrated in *Figure 5C* and *Figure 5—figure supplement 3*, where the free energy cost for tearing apart two nucleosomes with

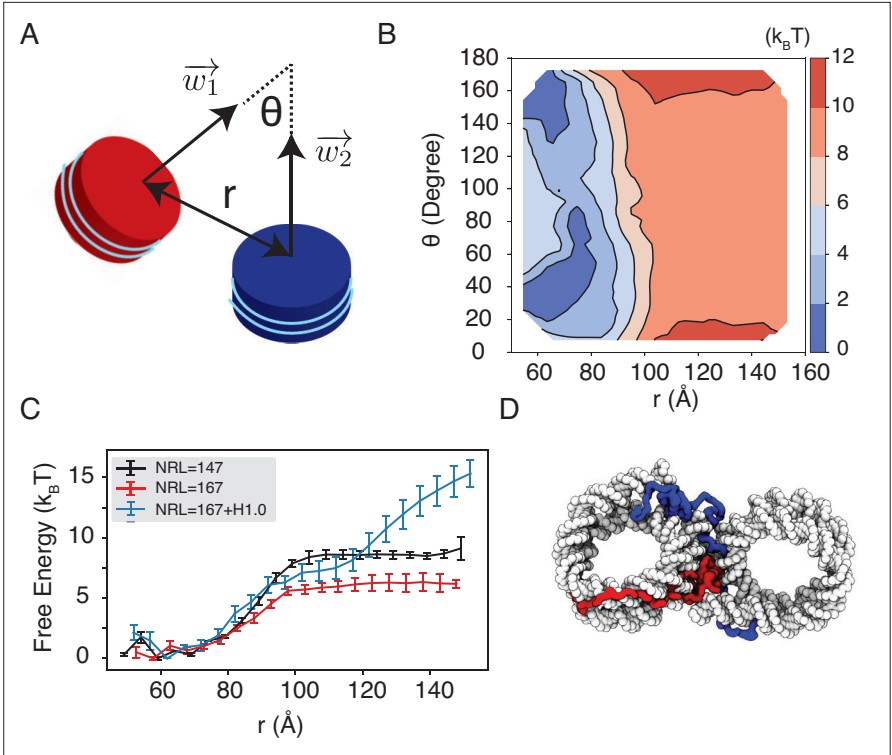

**Figure 5.** Simulations predict significant inter-nucleosome interactions at physiological conditions. (**A**) Illustration of the collective variable, $\theta$, defined as the angle between two nucleosomal planes, and $r$ defined as the distance between the nucleosome geometric centers. $\vec{w_1}$ and $\vec{w_2}$ represent the axes perpendicular to the nucleosomal planes. (**B**) The 2D binding free energy profile as a function of $\theta$ and $r$ at the physiological salt condition (150 mM NaCl and 2 mM MgCl$_2$) for nucleosomes with the 601 sequence. (**C**) Dependence of nucleosome binding free energy on nucleosome repeat length (NRL) and linker histone H1.0. Error bars were computed as the standard deviation of three independent estimates. (**D**) Representative structure showing linker histones (red and blue) mediating inter-nucleosomal contacts.

The online version of this article includes the following figure supplement(s) for figure 5:

**Figure supplement 1.** Dependence of inter-nucleosome interactions on the DNA sequence, related to *Figure 5* of the main text.

**Figure supplement 2.** The poly-dA:dT sequence produces a higher number of cross-nucleosome histone-DNA contacts compared to the poly-dG:dC sequence, related to *Figure 5* of the main text.

**Figure supplement 3.** Free energy profiles for the interactions between a pair of nucleosomes at different nucleosome repeat lengths (NRL) and in the presence of the linker histone H1.0, related to *Figure 5* of the main text.

167 bp DNA in the presence of linker histones (blue) is significantly higher than the curve for bare nucleosomes (red). Notably, at larger inter-nucleosome distances, the values even exceed those for 147 bp nucleosomes (black). A closer examination of the simulation configurations suggests that the disordered C-terminal tail of linker histones can extend and bind the DNA from the second nucleosome, thereby stabilizing the inter-nucleosomal contacts (as shown in *Figure 5D*). Our results are consistent with prior studies that underscore the importance of linker histones in chromatin compaction (*Finch and Klug, 1976*; *Zhou et al., 2021*), particularly in eukaryotic cells with longer linker DNA (*Routh et al., 2008*; *Dombrowski et al., 2022*).

## Discussion

We introduced a residue-level coarse-grained model with explicit ions to accurately account for electrostatic contributions to chromatin organization. The model achieves quantitative accuracy in reproducing experimental values for the binding affinity of protein-DNA complexes, the energetics

of nucleosomal DNA unwinding, nucleosome binding free energy, and the sedimentation coefficients of nucleosome arrays. It captures the counterion atmosphere around the nucleosome core particle as seen in all-atom simulations (*Materese et al., 2009*) and highlights the contribution of counterions to nucleosome stability. The coarse-grained model also succeeds in resolving the difference between monovalent and divalent ions, supporting the efficacy of divalent ions in neutralizing negative charges and offsetting repulsive interactions among DNA segments.

One significant finding from our study is the predicted strong inter-nucleosome interactions under the physiological salt environment, reaching approximately 9 $k_BT$. We showed that the much lower value reported in a previous DNA-origami experiment is due to the restricted nucleosomal orientation inherent to the device design. Unrestricted nucleosomes allow more close contacts to stabilize binding. A significant nucleosome binding free energy also agrees with the high forces found in single-molecule pulling experiments that are needed for chromatin unfolding (*Kruithof et al., 2009*; *Meng et al., 2015*; *Kaczmarczyk et al., 2020*). We also demonstrate that this strong inter-nucleosomal interaction is largely preserved at longer NRL in the presence of linker histone proteins. While post-translational modifications of histone proteins may influence inter-nucleosomal interactions, their effects are limited, as indicated by Ding et al. (*Ding et al., 2021*), and are unlikely to completely abolish the significant interactions reported here. Therefore, we anticipate that, in addition to molecular motors, chromatin regulators, and other molecules inside the nucleus, intrinsic inter-nucleosome interactions are important players in chromatin organization in vivo.

We focused our study on single chromatin chains. Strong inter-nucleosome interactions support the compaction and stacking of chromatin, promoting the formation of fibril-like structures. However, as shown in many studies (*Maeshima et al., 2016*; *Ricci et al., 2015*; *Ou et al., 2017*; *Zhang et al., 2022*), such fibril configurations can hardly be detected in vivo. It is worth emphasizing that this lack of fibril configurations does not contradict our conclusion on the significance of inter-nucleosome interactions. In a prior paper, we found that many in vivo factors, most notably crowding, could disrupt fibril configurations in favor of inter-chain contacts (*Liu et al., 2022*). The inter-chain contacts can indeed be driven by favorable inter-nucleosome interactions.

Several aspects of the coarse-grained model presented here can be further improved. For instance, the introduction of specific protein-DNA interactions could help address the differences in non-bonded interactions between amino acids and nucleotides beyond electrostatics (*Lin et al., 2021a*). Such a modification would enhance the model's accuracy in predicting interactions between chromatin and chromatin proteins. Additionally, the single-bead-per-amino-acid representation used in this study encounters challenges when attempting to capture the influence of histone modifications, which are known to be prevalent in native nucleosomes. Multiscale simulation approaches may be necessary (*Collepardo-Guevara et al., 2015*). One could first assess the impact of these modifications on the conformation of disordered histone tails using atomistic simulations. By incorporating these conformational changes into the coarse-grained model, systematic investigations of histone modifications on nucleosome interactions and chromatin organization can be conducted. Such a strategy may eventually enable the direct quantification of interactions among native nucleosomes and even the prediction of chromatin organization in vivo.

## Materials and methods
### Coarse-grained modeling of chromatin

The large system size of chromatin and the slow timescale for its conformational relaxation necessitates coarse-grained modeling. Following previous studies (*Leicher et al., 2020*; *Ding et al., 2021*; *Lin et al., 2021a*; *Lin et al., 2021b*; *Liu et al., 2022*), we adopted a residue-level coarse-grained model for efficient simulations of chromatin. The structure-based model (*Clementi et al., 2000*; *Noel et al., 2016*) was applied to represent the histone proteins with one bead per amino acid and to preserve the tertiary structure of the folded regions. The disordered histone tails were kept flexible without tertiary structure biases. A sequence-specific potential, in the form of the Lennard-Jones (LJ) potential and with the strength determined from the Miyazwa-Jernigan (MJ) potential (*Miyazawa and Jernigan, 1985*), was added to describe the interactions between amino acids. The 3SPN.2C model was adopted to represent each nucleotide with three beads and interactions among DNA beads follow the potential outlined in *Freeman et al., 2014*, except that the charge of each phosphate site

was switched from –0.6 to –1.0 to account for the presence of explicit ions. The Coulombic potential was applied between charged protein and DNA particles. In addition, a weak, non-specific LJ potential was used to account for the excluded volume effect among all protein-DNA beads. Detail expressions for protein-protein and protein-DNA interaction potentials can be found in *Ding et al., 2021*, and the Appendix section 'Coarse-grained protein-DNA model'.

We observe that residue-level coarse-grained models have been extensively utilized in prior studies to examine the free energy penalty associated with nucleosomal DNA unwinding (*Lequieu et al., 2016*; *Parsons and Zhang, 2019*; *Zhang et al., 2016*), sequence-dependent nucleosome sliding (*Lequieu et al., 2017*; *Brandani et al., 2018*), binding free energy between two nucleosomes (*Moller et al., 2019*), chromatin folding (*Ding et al., 2021*; *Liu et al., 2022*), the impact of histone modifications on tri-nucleosome structures (*Chang and Takada, 2016*), and protein-chromatin interactions (*Watanabe et al., 2018*; *Leicher et al., 2020*). The frequent quantitative agreement between simulation and experimental results supports the utility of such models in chromatin studies. Our introduction of explicit ions, as detailed in Appendix section 'Coarse-grained explicit ion model', further extends the applicability of these models to explore the dependence of chromatin conformations on salt concentrations.

## Coarse-grained modeling of counterions

Explicit particle-based representations for monovalent and divalent ions are needed to accurately account for electrostatic interactions (*Freeman et al., 2011*; *Hinckley and de Pablo, 2015*; *Hayes et al., 2015*; *Denesyuk and Thirumalai, 2015*; *Denesyuk et al., 2018*; *Wang et al., 2022*; *Sun et al., 2022*). We followed *Freeman et al., 2011*, to introduce explicit ions (see *Figure 1*) and adopted their potentials to describe the interactions between ions and nucleotide particles, with detailed expressions provided in the Appendix section 'Coarse-grained explicit ion model'. Parameters in these potentials were tuned by *Freeman et al., 2011*, to reproduce the radial distribution functions and the potential of mean force between ion pairs determined from all-atom simulations.

**Table 1.** Summary of parameters used to describe interactions between ions and charged particles. See text section 'Coarse-grained explicit ion model' for definitions of various parameters.

| Coarse-grained pair | $\epsilon$(kcal/mol) | $\sigma$(Å) | $r_{m\epsilon}$(Å) | $\sigma_\epsilon$(Å) | $H_1$(kcal/mol) | $r_{mh1}$(Å) | $\sigma_{h1}$(Å) | $H_2$(kcal/mol) | $r_{mh2}$(Å) | $\sigma_{h2}$(Å) |
|---|---|---|---|---|---|---|---|---|---|---|
| P-P | 0.18379 | 6.86 | 6.86 | 0.5 | – | – | – | – | – | – |
| Na$^+$-P | 0.02510 | 4.14 | 3.44 | 1.25 | 3.15488 | 4.1 | 0.57 | 0.47801 | 6.5 | 0.4 |
| Na$^+$-AA$^{+*}$ | 0.239 | 4.065 | 3.44 | 1.25 | 3.15488 | 4.1 | 0.57 | – | – | – |
| Na$^+$-AA$^{-\dagger}$ | 0.239 | 4.065 | 3.44 | 1.25 | 3.15488 | 4.1 | 0.57 | 0.47801 | 6.5 | 0.4 |
| Mg$^{2+}$-P | 0.1195 | 4.87 | 3.75 | 1.0 | 1.29063 | 6.1 | 0.5 | 0.97992 | 8.3 | 1.2 |
| Mg$^{2+}$-AA$^+$ | 0.239 | 3.556 | 3.75 | 1.0 | 1.29063 | 6.1 | 0.5 | – | – | – |
| Mg$^{2+}$-AA$^-$ | 0.239 | 3.556 | 3.75 | 1.0 | 1.29063 | 6.1 | 0.5 | 0.97992 | 8.3 | 1.2 |
| Cl$^-$-P | 0.08121 | 5.5425 | 4.2 | 0.5 | 0.83652 | 6.7 | 1.5 | – | – | – |
| Cl$^-$-AA$^+$ | 0.239 | 4.8725 | 4.2 | 0.5 | 0.83652 | 6.7 | 1.5 | 0.47801 | 5.6 | 0.4 |
| Cl$^-$-AA$^-$ | 0.239 | 4.8725 | 4.2 | 0.5 | 0.83652 | 6.7 | 1.5 | – | – | – |
| Na$^+$-Na$^+$ | 0.01121 | 2.43 | 2.7 | 0.57 | 0.17925 | 5.8 | 0.57 | – | – | – |
| Na$^+$-Mg$^{2+}$ | 0.04971 | 2.37 | 2.37 | 0.5 | – | – | – | – | – | – |
| Na$^+$-Cl$^-$ | 0.08387 | 3.1352 | 3.9 | 2.06 | 5.49713 | 3.3 | 0.57 | 0.47801 | 5.6 | 0.4 |
| Mg$^{2+}$-Mg$^{2+}$ | 0.89460 | 1.412 | 1.412 | 0.5 | – | – | – | – | – | – |
| Mg$^{2+}$-Cl$^-$ | 0.49737 | 4.74 | 4.48 | 0.57 | 1.09943 | 5.48 | 0.44 | 0.05975 | 8.16 | 0.35 |
| Cl$^-$-Cl$^-$ | 0.03585 | 4.045 | 4.2 | 0.56 | 0.23901 | 6.2 | 0.5 | – | – | – |

*Positive amino acids.
†Negative amino acids.

**Table 2.** Summary of parameters used to describe the WCA interactions between ions and neutral particles.

See text section 'Coarse-grained explicit ion model' for definitions of various parameters.

| Coarse-grained pair | $\epsilon$(kcal/mol) | $\sigma$(Å) |
|---|---|---|
| Na$^+$-S* | 0.239 | 4.315 |
| Na$^+$-A† | 0.239 | 3.915 |
| Na$^+$-T‡ | 0.239 | 4.765 |
| Na$^+$-G§ | 0.239 | 3.665 |
| Na$^+$-C¶ | 0.239 | 4.415 |
| Na$^+$-AA** | 0.239 | 4.065 |
| Mg$^{2+}$-S | 0.239 | 3.806 |
| Mg$^{2+}$-A | 0.239 | 3.406 |
| Mg$^{2+}$-T | 0.239 | 4.256 |
| Mg$^{2+}$-G | 0.239 | 3.156 |
| Mg$^{2+}$-C | 0.239 | 3.906 |
| Mg$^{2+}$-AA** | 0.239 | 3.556 |
| Cl$^-$-S | 0.239 | 5.1225 |
| Cl$^-$-A | 0.239 | 4.7225 |
| Cl$^-$-T | 0.239 | 5.5725 |
| Cl$^-$-G | 0.239 | 4.4725 |
| Cl$^-$-C | 0.239 | 5.2225 |
| Cl$^-$-AA** | 0.239 | 4.8725 |

*Sugar.
†Adenine base.
‡Thymine base.
§Guanine base.
¶Cytosine base.
**Non-charged amino acids.

This explicit ion model was originally introduced for nucleic acid simulations. We generalized the model for protein simulations by approximating the interactions between charged amino acids and ions with parameters tuned for phosphate sites. Parameter values for ion-amino acid interactions are provided in *Table 1* and *Table 2*.

## Details of molecular dynamics simulations

We simulated various chromatin systems, including a single-nucleosome, two-nucleosomes, and a 12-mer nucleosome array. The initial configurations for the molecular dynamics simulations were constructed based on the crystal structure of a single nucleosome with PDB ID: 1KX5 (*Davey et al., 2002*) and 3LZ1 (*Vasudevan et al., 2010*), or a tetranucleosome with PDB ID: 1ZBB (*Schalch et al., 2005*). We used the 3DNA software (*Lu and Olson, 2003*) to add additional DNA, connect and align nucleosomes, and extend the chain length as necessary. Further details on constructing the initial configurations are provided in the Appendix section 'Ionic dependence of the conformation for a 12-mer nucleosomal array'. Chromatin was positioned at the center of a cubic box with a length selected to avoid interactions between nucleosomes and their periodic images. Counterions were added on a uniformly spaced grid to achieve the desired salt concentrations and neutralize the system. The number of ions and the size of simulation boxes are provided in *Table 3*.

All simulations were performed at constant temperature and constant volume (NVT) using the software package LAMMPS (*Plimpton, 1995*). The electrostatic interactions were implemented with

**Table 3.** Summary of simulation setups used in this study.
Additional simulation details can be found in text section 'Molecular dynamics simulation details'.

| Studies | Box size (nm³) | Number of Na⁺ | Number of Mg²⁺ | Number of Cl⁻ |
|---|---|---|---|---|
| Single nucleosome 100 mM NaCl+0.5 mM MgCl$_2$ | 216,000 | 13,017 | 65 | 13,003 |
| Twelve nucleosomes 5 mM NaCl | 1,331,000 | 6196 | 0 | 4006 |
| Twelve nucleosomes 150 mM NaCl | 216,000 | 21,695 | 0 | 19,505 |
| Twelve nucleosomes 0.6 mM MgCl$_2$ | 3,375,000 | 0 | 2314 | 2438 |
| Twelve nucleosomes 1 mM MgCl$_2$ | 3,375,000 | 0 | 3127 | 4064 |
| Two 147 bp 601-seq nucleosomes 35 mM NaCl+11 mM MgCl$_2$ | 125,000 | 2922 | 828 | 4290 |
| Two 147 bp 601-seq nucleosomes 150 mM NaCl+2 mM MgCl$_2$ | 216,000 | 19,505 | 260 | 19,737 |
| Two 147 bp poly-dA:dT nucleosomes 150 mM NaCl+2 mM MgCl$_2$ | 216,000 | 19,505 | 260 | 19,737 |
| Two 147 bp poly-dG:dC nucleosomes 150 mM NaCl+2 mM MgCl$_2$ | 216,000 | 19,505 | 260 | 19,737 |
| Two 167 bp 601-seq nucleosomes 150 mM NaCl+2 mM MgCl$_2$ | 216,000 | 19,505 | 260 | 19,657 |
| Two 167 bp 601-seq nucleosomes with H1.0 150 mM NaCl+2 mM MgCl$_2$ | 216,000 | 19,505 | 260 | 19,763 |

the particle-particle particle-mesh solver, with the relative root-mean-square error in per-atom force set to 0.0001 (*Hockney and Eastwood, 2021*). A Nosé-Hoover style algorithm (*Shinoda et al., 2004*) was used to maintain the system temperature at 300 K with a damping parameter of 1 ps. We further modeled the histone core and the inner layer of the nucleosomal DNA together as a rigid body to improve computational efficiency. This approximation does not affect the thermodynamic properties of chromatin (*Ding et al., 2021*; *Liu et al., 2022*). Umbrella simulations were used to enhance the sampling of the conformational space (*Torrie and Valleau, 1977*), and details of the collective variables employed in these simulations are provided in the Appendix section 'Molecular dynamics simulation details'. All the results presented in the main text are reweighted from the biased simulations by the weighted histogram algorithm (*Kumar et al., 1992*).

## Acknowledgements

This work was supported by the National Institutes of Health (Grant R35GM133580) and the National Science Foundation (Grant MCB-2042362).

## Additional information

### Competing interests

Bin Zhang: Reviewing editor, *eLife*. The other author declares that no competing interests exist.

### Funding

| Funder | Grant reference number | Author |
|---|---|---|
| National Institute of General Medical Sciences | R35GM133580 | Bin Zhang |
| National Science Foundation | MCB-2042362 | Bin Zhang |

The funders had no role in study design, data collection and interpretation, or the decision to submit the work for publication.

### Author contributions

Xingcheng Lin, Conceptualization, Software, Formal analysis, Validation, Investigation, Visualization, Methodology, Writing – original draft, Writing – review and editing; Bin Zhang, Conceptualization,

Software, Formal analysis, Supervision, Funding acquisition, Validation, Investigation, Visualization, Methodology, Writing – original draft, Project administration, Writing – review and editing

**Author ORCIDs**
Xingcheng Lin ⬥ https://orcid.org/0000-0002-9378-6174
Bin Zhang ⬥ http://orcid.org/0000-0002-3685-7503

Joint public review: https://doi.org/10.7554/eLife.90073.3.sa1
Author response https://doi.org/10.7554/eLife.90073.3.sa2

## Additional files

### Supplementary files
• MDAR checklist

### Data availability

The current manuscript is a computational study, so no data have been generated for this manuscript. Modelling code is uploaded to GitHub: https://github.com/ZhangGroup-MITChemistry/Explicit_Ion_Chromatin (copy archived at *Lin, 2023*).

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

## Appendix 1

### Coarse-grained protein-DNA model

The force fields describing protein-protein, protein-DNA, and DNA-DNA interactions followed previous studies (*Ding et al., 2021*; *Liu et al., 2022*; *Freeman et al., 2014*; *Zhang et al., 2016*). Detailed expressions for the potential energies can be found in these references. Specifically, for DNA-DNA interactions, we followed the approach described in *Freeman et al., 2011*, and with the parameters updated to the latest version of the DNA model 3SPN.2C (*Freeman et al., 2014*).

Protein-protein interactions include structure-based terms extracted from the initial configuration and generic terms for specific amino acid interactions. We first generated the bonded and non-bonded structure-based interactions within histone proteins using the Shadow algorithm (*Noel et al., 2012*) implemented by the SMOG software package (*Noel et al., 2016*). We further scaled the non-bonded interaction strength (*Noel et al., 2016*) by 2.5 to prevent proteins from unfolding at 300 K. Interactions between histone proteins from different nucleosomes were described using the MJ potential (*Miyazawa and Jernigan, 1985*) scaled by a factor of 0.4. We have shown in our previous studies that the scaled MJ potential gives a balanced modeling of the radius of gyration for both ordered and disordered proteins (*Ding et al., 2021*).

Protein-DNA interactions include Coulombic interactions and the excluded volume effect. Unlike the previous Debye-Hückel treatment of electrostatic interactions in an implicit-solvent environment, we modeled the electrostatics between proteins and DNA using the Coulombic potential

$$U_{\text{elec}} = \frac{1}{4\pi\epsilon_0} \frac{q_i q_j}{\epsilon_0 r}, \tag{1}$$

where $\epsilon_0 = 78.0$ is the dielectric constant of the bulk solvent. $q_i$ and $q_j$ correspond to the charges of the two particles. The excluded volume effect was modeled using the WCA potential with the following form

$$U_{\text{WCA}} = \begin{cases} 4\epsilon^{\text{PD}}[(\frac{\sigma}{r})^{12} - (\frac{\sigma}{r})^6] + \epsilon^{\text{PD}} & r < r_{\text{cut}} \\ 0 & r > r_{\text{cut}}. \end{cases} \tag{2}$$

The cutoff distance $r_{\text{cut}}$ was set to $2^{\frac{1}{6}}\sigma$, with $\sigma = 5.7$ Å. The interactive strength $\epsilon^{\text{PD}}$ was set as 0.02987572 kcal/mol. More details of this potential can be found in *Zhang et al., 2016*.

### Coarse-grained explicit ion model

Following *Freeman et al., 2011*, we adopted three terms to describe interactions between charged particles and ions: the Coulombic potential for electrostatic interactions, the Gaussian potential for the hydration effect, and the LJ potential for the excluded volume effect. Thus,

$$U = U_{\text{elec}} + U_{\text{hydr}} + U_{\text{LJ}}. \tag{3}$$

The electrostatic potential

$$U_{\text{elec}} = \frac{1}{4\pi\epsilon_0} \frac{q_i q_j}{\epsilon_{\text{D}}(r) r}, \tag{4}$$

where $\epsilon_{\text{D}}(r)$ is a distance-dependent dielectric constant given in the form

$$\epsilon_{\text{D}}(r) = (\frac{5.2 + \epsilon_s}{2}) + (\frac{\epsilon_s - 5.2}{2}) \tanh[\frac{r - r_{\text{m}\epsilon}}{\sigma_\epsilon}]. \tag{5}$$

$\epsilon_s = 78.0$ is the dielectric constant of the bulk solvent, and values for $\sigma_\epsilon$ and $r_{m\epsilon}$ are provided in *Table 1*. The distance cutoff of this potential is set at 20.0 Å. Electrostatic interactions outside this cutoff are computed in reciprocal space.

The hydration potential

$$U_{\text{hydr}} = \frac{H}{\sigma_h \sqrt{2\pi}} \exp[-\frac{(r - r_{mh})^2}{2\sigma_h^2}]. \tag{6}$$

$r_{mh}$, $\sigma_h$, and $H$ represent the midpoint, the width, and the height of the hydration shell, respectively, and their ion-specific values are provided in *Table 1*. Any pair of ions experiences one hydration potential defined above. For pairs formed with ions of distinct types, a second hydration potential with a different set of parameters is applied, with parameters provided in *Table 1*. The distance cutoff of this potential is set at 12.0 Å.

The LJ potential is given by

$$U_{\text{LJ}} = 4\epsilon[(\frac{\sigma}{r})^{12} - (\frac{\sigma}{r})^{6}]. \tag{7}$$

The ion-specific values of $\epsilon$ and $\sigma$ are given in *Table 1*, and the distance cutoff of this potential is set at 12.0 Å.

The interactions between neutral particles and ions are described by the WCA potential

$$U_{\text{excl}} = \begin{cases} 4\epsilon_{\text{excl}}[(\frac{\sigma}{r})^{12} - (\frac{\sigma}{r})^{6}] + \epsilon_{\text{excl}} & r < r_{\text{cut}} \\ 0 & r > r_{\text{cut}}. \end{cases} \tag{8}$$

$r_{\text{cut}} = 2^{\frac{1}{6}}\sigma$ is located at the minimum of the corresponding LJ potential. Values for $\sigma$ and $\epsilon_{\text{excl}}$ follow the parameters given by *Freeman et al., 2011*, and are presented in *Table 2*.

## Molecular dynamics simulation details

All simulations were carried out using the software Lammps (*Plimpton, 1995*) with the force fields defined in the previous two sections. Umbrella sampling simulations (*Torrie and Valleau, 1977*) were performed using the Plumed software package (*The PLUMED consortium, 2019*). We used the weighted histogram analysis method (*Kumar et al., 1992*) implemented by the SMOG software package (*Noel et al., 2016*) to process the simulation data and compute the free energy profiles.

## Binding free energy of protein-DNA complexes

We carried out a series of umbrella-sampling simulations to compute the binding free energies of a set of nine protein-DNA complexes with experimentally documented binding dissociation constants (*Dragan et al., 2003a*; *Dragan et al., 2004*; *Dragan et al., 2003b*; *Dragan et al., 2006*; *Privalov et al., 2011*). Initial configurations of these simulations were prepared using the crystal structures with the corresponding PDB IDs listed in *Figure 2—figure supplement 1*.

The simulations were performed under the same experimental conditions of 100 mM monovalent ions. We used a spring constant of 0.01 kcal/mol/Å$^2$ to restrain the distance between the geometric centers of protein and DNA. The centers of the umbrella windows were placed on a uniform grid of [0.0:140.0:10.0] Å, and each umbrella trajectory lasts for 7.15 million steps, with a time step of 2.0 fs. We excluded the first 3 million steps when constructing the free energy profile.

## Single-nucleosome simulations for DNA unwinding energetics

To study DNA unwinding from a 601-sequence nucleosome, we built the system by combining histone proteins with explicit coordinates for the disordered tails from PDB ID: 1KX5 (*Davey et al., 2002*) with the DNA structure from PDB ID: 3LZ1 (*Vasudevan et al., 2010*).

Umbrella simulations with the DNA end-to-end distance as the collective variable was performed to determine the free energy profile. The end-to-end distance was defined as the geometric center distance between the first and last five base pairs. We used a harmonic umbrella potential with a spring constant of 0.001 kcal/(mol·Å$^2$), and the umbrella centers were placed on a uniform grid of [30.0:510.0:30.0] Å. To increase computational efficiency, the histone core proteins and the two nucleotides located on the dyad axis of the nucleosome were rigidified during the simulations. Each umbrella trajectory lasts for 13.65 million steps, with a time step of 10 fs, and we excluded the first 3 million steps when constructing the free energy profile.

The simulation used the same ionic concentration as the experiment, which includes 0.10 M NaCl and 0.5 mM MgCl$_2$ (*Hall et al., 2009*). The cubic simulation box size was set to 600 Å. Extra ions were added to neutralize the system. In total, the system includes a total of 13,017 Na$^+$, 65 Mg$^{2+}$, and 13,003 Cl$^-$ ions.

## Ionic dependence of the conformation for a 12-mer nucleosomal array

We constructed a nucleosomal array of 12 nucleosomes with 20 bp linker DNA to study the impact of different ions on the higher-order chromatin organization. Using the protocol outlined in a previous study (*Liu et al., 2022*), we started with a nucleosome unit extracted from the tetranucleosome crystal structure (PDB ID: 1ZBB) (*Schalch et al., 2005*). To connect multiple nucleosome units, we left an extra 20 bp linker DNA at the exit site of the nucleosome. We connected 12 nucleosome units to build the 12-mer nucleosomal array, with an additional 20 bp linker DNA at the end. This 20 bp extra linker of the last nucleosome unit was removed to complete the system setup. To ensure complete histone proteins with disordered tails, we replaced the histone proteins of the nucleosome units with those from the crystal structure with PDB ID: 1KX5 (*Davey et al., 2002*).

To enhance conformational sampling, we performed umbrella simulations with the collective variable $Q$ defined as

$$Q = \frac{1}{N} \sum_{i}^{N} \exp\left(-\frac{(r_i - d_0^i)^2}{2r_0^2}\right) \tag{9}$$

$i$ enumerates all the nucleosome pairs in the system and $r_i$ is the distance between the $i$th pair. $N=66$ is the number of nucleosome pairs, and $r_0 = 20.0$ Å. $d_0^i$ corresponds to the distance between the $i$th pair of nucleosomes determined from the reference two-start structure. $Q$ measures the similarity of a given 12-mer configuration to the reference two-start structure, with larger values representing higher similarity. The reference two-start fibril structure was built in our previous study (*Liu et al., 2022*) by aligning the structure with a template generated by the software fiberModel (*Koslover et al., 2010*). We placed the umbrella centers at [0.40:0.90,0.1] and used a spring constant of 50.0 kcal/mol in the harmonic potentials. Each umbrella trajectory lasted 7.8 million steps, with a time step of 10 fs. We used the last 4.8 million steps to construct free energy profiles and compute ensemble averages.

We simulated the nucleosome array under four ionic conditions for comparison with experimental measurements (*Correll et al., 2012*). The 12-mer was placed in the center of a cubic box, and counterions were added to reach the desired concentration. Excess ions were also introduced to ensure the net neutrality of the system. Specific values of the simulation box size and counterion numbers are provided in *Table 3* for reference.

## Binding free energy between nucleosomes

### Simulations at high salt concentrations

To determine the inter-nucleosome interactions and compare them with the DNA-origami-based experiment (*Funke et al., 2016*), we simulated a pair of 601-sequence nucleosomes. The nucleosomes were built in the same way as the single-nucleosome DNA unwrapping simulation. The simulation box size was 500 Å, and extra ions were added to neutralize the system. The system comprised a total of 2922 $Na^+$, 828 $Mg^{2+}$, and 4290 $Cl^-$ ions.

We defined the internal coordinate system for each nucleosome to control their relative orientations as follows. The center of each nucleosome was determined using the geometric center of the list of residues: 63–120, 165–217, 263–324, 398–462, 550–607, 652–704, 750–811, and 885–949. The IDs continuously index residues from chain A to chain H of the crystal structure with PDB ID: 1KX5. To define the nucleosome plane, we chose another point based on the geometric center of the nucleosome dyad site (residue ID: 81–131, 568–618). The unit vector pointing from the nucleosome center to the dyad site center is denoted as $\vec{v}$. We further introduced another unit vector $\vec{u}$ in the nucleosome plane that points from the nucleosome center to a point defined as the geometric center of residues 63–120, 165–217, 750–811, and 885–949. Finally, the unit vector perpendicular to the nucleosome plane, $\vec{w}$, is determined as the cross product $\vec{u} \times \vec{v}$. We use $\vec{w}_1$ and $\vec{w}_2$ to differentiate the two nucleosomes.

We utilized two collective variables for the system without constraints to perform the umbrella simulations. The first collective variable measures the distance $r$ between the two nucleosome centers. The second collective variable corresponds to the angle $\theta$ between the two unit vectors, $\vec{w}_1$ and $\vec{w}_2$. The umbrella centers were placed on a uniform grid of [60.0, 130.0, 10.0] Å × [0.0, 180.0, 30.0] degrees. The spring constants of the harmonic potentials are 0.01 kcal/(mol · Å$^2$) and 0.001 kcal/(mol · $\circ^2$).

For the system that mimics the DNA-origami experiments, we imposed several spatial restraints such that the nucleosomes unbind along a specific pathway (*Figure 4—figure supplement 1*). As detailed below, the restraints ensure that the first nucleosome is fixed on the *X-Y* plane while the second nucleosome moves along an arc 15 nm away from the origin.

1. We introduced three virtual sites, denoted as $O, A,$ and $B$, with Cartesian coordinates as [0, 0, 0], [150, 0, 0], and [0, 150, 0] Å, respectively. The vectors $\overrightarrow{OA}$ and $\overrightarrow{OB}$ define the *X-Y* plane. We further denote the centers of the two nucleosomes as $C_1$ and $C_2$.
2. The first nucleosome was restrained at site $B$ using a harmonic potential with a spring constant of 100 kcal/(mol · Å²). In addition, to mimic its attachment to the bottom arm of the DNA origami, we forced this nucleosome to be parallel to the *X-Y* plane. Specifically, we restrained the angles between $\vec{w}_1$ and $\overrightarrow{OA}$ or $\overrightarrow{OB}$ to be 90°. The spring constant of these harmonic restraints was set to 100 kcal/(mol · ○²).
3. To mimic the attachment of the second nucleosome to the upper arm of the origami, we restrained the distance between $C_2$ and site $O$ as 150 Å with a harmonic potential. The spring constant of this potential was set to 100.0 kcal/(mol · Å²). In addition, we ensured that the second nucleosome is parallel to the plane formed by the vector $\overrightarrow{OA}$ and the vector connecting site $O$ to $C_2$. Two harmonic potentials were applied on the angles between $\vec{w}_2$ and $\overrightarrow{OA}$ or $\overrightarrow{OC_2}$ to restrict them to 90°. The spring constant of these restraints was set to $100 \, \text{kcal/(mol} \cdot ○^2)$. We further restricted the second nucleosome to only move in the *Y-Z* plane by biasing the angle between $\overrightarrow{OC_2}$ and $\overrightarrow{OA}$ to 90° with a spring constant of 100 kcal/(mol · ○²).
4. Finally, we ensured that the dyad axes of the two nucleosomes in our system are at an angle of 78°, as done experimentally (*Funke et al., 2016*), by applying a harmonic potential on the angle between the $\vec{v}$ of the two nucleosomes. The spring constant of this potential was set to 100 kcal/(mol · ○²).

We used the angle $\theta$ between the two vectors $\vec{w}_1$ and $\vec{w}_2$ as the collective variable for umbrella simulations. The umbrella centers were placed on a uniform grid of [0.0, 110.0, 5.0] degrees. The spring constant of the harmonic potential was set as 0.01 kcal/(mol · ○²). Each umbrella simulation lasted 13 million steps and a time step of 10 fs. The first 3 million steps of the simulation were discarded as equilibration.

## Simulations at the physiological salt concentration

We performed a series of simulations under the physiological salt concentration, i.e., 150 mM NaCl and 2 mM MgCl$_2$ for nucleosomes with different repeat lengths and DNA sequences. The 601-sequence nucleosomes were built in the same way as the single-nucleosome simulations. To explore the effect of DNA sequences on inter-nucleosomal interactions, we replaced the original nucleosomal DNA with poly-dA:dT and poly-dG:dC sequences.

To investigate the effect of linker DNA, we simulated nucleosomes with a repeat length of 167 bp. We added 10 base pairs of poly-dA:dT sequences on each side of the existing 147 bp 601-nucleosomal DNA using the software X3DNA (*Lu and Olson, 2003*). Specifically, we generated an 11-base-pair linker DNA of poly-dA:dT sequence. The additional DNA base pair was created to align the linker DNA with the existing nucleosomal DNA. This alignment was performed such that this additional base pair overlapped with the nucleosomal DNA's first or last base pair, fixing the linker DNA's orientation. Finally, we deleted the additional base pair of DNA after the alignment.

Additionally, we built nucleosomes with linker histones using a recently resolved chromatosome structure through cryoelectron microscopy (*Zhou et al., 2021*). The experimentally determined structure (PDB ID: 7K5X) includes a 197 bp 601-sequence nucleosome with the globular domain of H1.0. As the disordered regions of the linker histone were not resolved in the structure, we modeled them based on the protein sequence using the software Modeller (*Eswar et al., 2006*) and connected the modeled structure to the globular domain. We then replaced the histone proteins with those from the PDB ID: 1KX5 to provide explicit coordinates for the histone tails. Only the central 167 bp of DNA was kept to build a system with 10 bp linker DNA. The globular domain of H1.0 was bound to the nucleosome dyad and simulated with the histone core protein as a rigid body for computational efficiency.

The numbers of ions and box sizes in each simulation are provided in *Table 3*. We employed the same two collective variables as the unrestrained simulation at high salt concentrations to conduct umbrella simulations. The umbrella centers were placed on a uniform grid of [60.0, 130.0, 10.0] Å × [0.0, 180.0, 30.0] degrees. The spring constants of the harmonic potentials are 0.01 kcal/(mol · Å²) and 0.001 kcal/(mol · ∘²). Each simulation lasted 13 million steps with a time step of 10 fs. We excluded the first 3 million steps when constructing the free energy profiles.

## Details of simulation analysis

### Number of ions bound to DNA and histone proteins

To calculate the number of ions bound to the nucleosomal DNA and histone proteins, we used the COORDINATIONNUMBER command available in the Plumed (*The PLUMED consortium, 2019*) software package.

For example, for every $Na^+$, we computed the coordinate number as $CN = \sum_i s(r_i)$, where $i$ loops over all coarse-grained DNA sites and $r_i$ is the distance between the ion and the $i$th DNA bead. $s(r)$ is a switching function defined as

$$s(r) = \frac{1 - (\frac{r - d_0}{r_0})^n}{1 - (\frac{r - d_0}{r_0})^m},$$ (10)

where $d_0 = 0.0$, $r_0 = 10.0$, $n = 15$, and $m = 30$. An ion with a coordination number greater than 1 was considered bound to DNA. We followed the same procedure to calculate the number of ions bound to histone proteins.

The calculations were performed using the open-source, community-developed PLUMED library (*The PLUMED consortium, 2019*), version 2.4 (*Tribello et al., 2014*; *The PLUMED consortium, 2019*).

### Number of unwrapped DNA base pairs

We computed the number of unwrapped DNA base pairs using a similar procedure to the one used for calculating the number of bound ions.

First, we computed a coordination number for each DNA base pair to determine whether it was bound to the histone core. The coordinate number was defined as $CN = \sum_i \sum_j s(r_{i,j})$, where $i$ loops over all coarse-grained sites of the corresponding DNA base pair and $j$ loops over all coarse-grained sites of the histone core. $s(r)$ is defined in *Equation 10* with $d_0 = 0.0$, $r_0 = 8.0$, $n = 15$, and $m = 30$. A DNA base pair with CN greater than 1 was considered bound to histone proteins. As the histone core is not a perfect cylinder, there were several continuous regions of bound DNA interspersed by regions of unbound DNA. To avoid ambiguity, we defined the wrapped base pairs, $N_{wrapped}$, as those between the first and last bound base pairs. Correspondingly, the number of unwrapped base pairs was $N_{unwrapped} = 147 - N_{wrapped}$.

We set $r_0 = 8.0$ Å when computing the switching function. At larger values for $r_0$, we found that the calculated numbers overestimate the unwrapped base pairs, as seen from visual inspection of the structures (*Figure 2*, *Appendix 1—figure 1*).

### Sedimentation coefficients of nucleosome arrays

We calculated the sedimentation coefficients for the 12-mer nucleosome array using the HullRad method (*Fleming and Fleming, 2018*) with the following equation

$$s = 10^8 (\frac{M - M\bar{v}\rho_{20,w}}{N_A 6\pi\eta_0 R_T}).$$ (11)

$M$ is the molar mass of the molecule, $N_A$ is Avogadro's number, and $\bar{v}$ is the partial specific volume. $\rho_{20,w}$ is the density of water at 20°C, and $\eta_0$ is the water viscosity at 20°C. $R_T$ is the translational hydrodynamic radius calculated based on the convex hull of the target biomolecule.

### Estimation of error bars

We estimated the error bars of the 12-mer simulations based on the standard deviation calculated from the probability distribution of the variables (*Figure 3—figure supplement 1*), i.e.,

$$\sigma(X) = \sqrt{E[X^2] - (E[X])^2} \tag{12}$$

where $E[X]$ is the expected value of $X$.

We divided the trajectories into three equal-length partitions for all the other simulations and computed the free energy profiles independently. The error bars were estimated as the standard deviation of the three independent estimates.

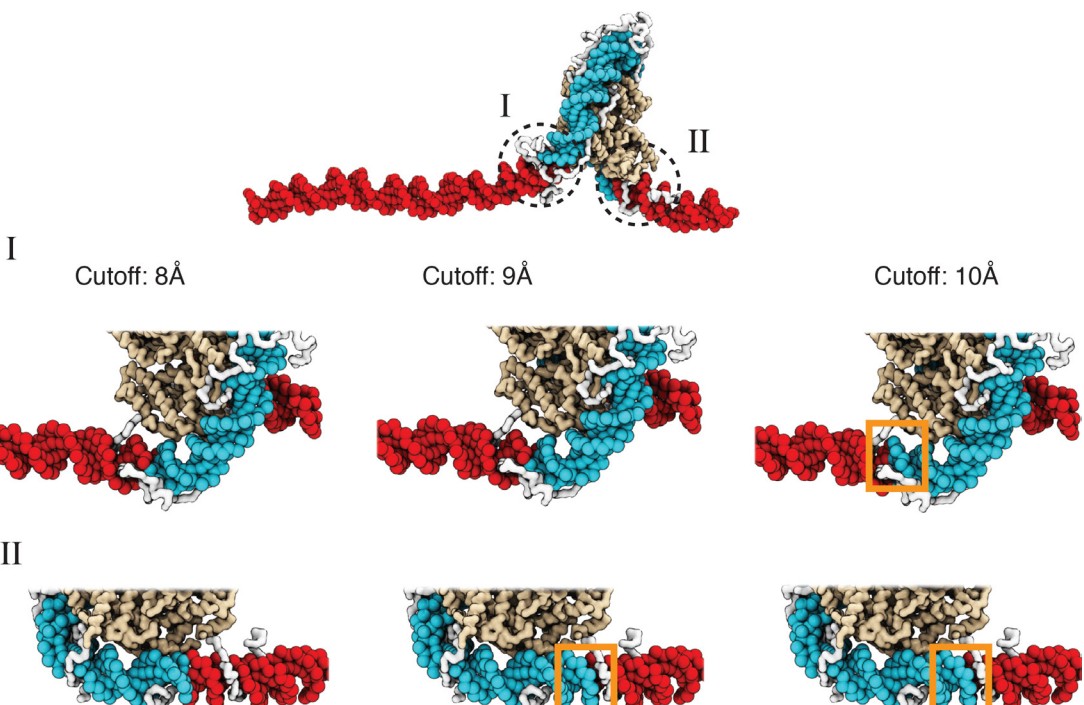

**Appendix 1—figure 1.** A cutoff value of 8.0 Å produces more accurate values for the number of unwrapped DNA base pairs as determined from visual inspection of representative configurations, related to *Figure 2* of the main text. See text section 'Number of unwrapped DNA base pairs' for additional discussions. A typical nucleosome structure with most of the outer layer DNA unwrapped was used to examine the impact of different cutoff values. The histone core is colored in gold, with histone tails in white, the wrapped DNA in blue, and the unwrapped DNA in red. The discrepancy among various cutoff values is evident in the highlight regions enclosed by dotted circles. As shown in the zoom-ins in the middle panel, a cutoff of 10 Å results in three additional base pairs of DNA detected as wrapped in I (highlighted in orange square). However, these extra base pairs not detected with a cutoff of 8 Å are visibly detached from histone proteins. Similarly, 9 Å and 10 Å cutoff values result in five extra base pairs of DNA detected as wrapped in II (highlighted in orange squares in the bottom panel).

