## [Editor Report · eLife assessment]

The authors have developed a **compelling** coarse-grained simulation approach for nucleosome-nucleosome interactions within a chromatin array. The data presented are **solid** and provide new insights that allow for predictions of how chromatin interactions might occur in vivo. The tools presented herein will be **valuable** for the chromosome biology field.

---

## [Referee Report · Joint public review]

In this manuscript, the authors introduced an explicit ion model using the coarse-grained modelling approach to model the interactions between nucleosomes and evaluate their effects on chromatin organization. The strength of this method lies in the explicit representation of counterions, especially divalent ions, which are notoriously difficult to model. To achieve their aims and validate the accuracy of the model, the authors conducted coarse-grained molecular dynamics simulations and compared predicted values to the experimental values of the binding energies of protein-DNA complexes and the free energy profile of nucleosomal DNA unwinding and inter-nucleosome binding. Additionally, the authors employed umbrella sampling simulations to further validate their model, reproducing experimentally measured sedimentation coefficients of chromatin under varying salt concentrations of monovalent and divalent ions.

The significance of this study lies in the authors' coarse-grained model which can efficiently capture the conformational sampling of molecules while maintaining a low computational cost. The model reproduces the scale and, in some cases, the shape of the experimental free energy profile for specific molecule interactions, particularly inter-nucleosome interactions. Additionally, the authors' method resolves certain experimental discrepancies related to determining the strength of inter-nucleosomal interactions. Furthermore, the results from this study support the crucial role of intrinsic physicochemical interactions in governing chromatin organization within the nucleus.

The authors have successfully addressed the majority of my key concerns. I appreciate the clarification regarding the parameterization from Pablo's lab and the addition of comparisons of energy profiles as a function of inter-nucleosome distances.

However, the statement "The agreement is evident" may not sufficiently capture the essence of Figure S4, as there is a shortage of substantial agreement. The authors rightly acknowledge it but should delineate the nature of the observed discrepancies.

---

## [Author Response]

The following is the authors’ response to the original reviews.

**eLife assessment**
The authors have developed a compelling coarse-grained simulation approach for nucleosome-nucleosome interactions within a chromatin array. The data presented are solid and provide new insights that allow for predictions of how chromatin interactions might occur in vivo, but some of the claims should be tempered. The tools will be valuable for the chromosome biology field.

Response: We want to thank the editors and all the reviewers for their insightful comments. We have made substantial changes to the manuscript to improve its clarity and temper necessary claims, as detailed in the responses, and we performed additional analyses to address the reviewers’ concerns. We believe that we have successfully addressed all the comments, and the quality of our paper has improved significantly.

In the following, we provide point-to-point responses to all the reviewer comments.

**RESPONSE TO REFEREE 1:**
Comment 0: This study develops and applies a coarse-grained model for nucleosomes with explicit ions. The authors perform several measurements to explore the utility of a coarse-grained simulation method to model nucleosomes and nucleosome arrays with explicit ions and implicit water. ’Explicit ions’ means that the charged ions are modeled as particles in simulation, allowing the distributions and dynamics of ions to be measured. Since nucleosomes are highly charged and modulated by charge modifications, this innovation is particularly relevant for chromatin simulation.

Response: We thank the reviewer’s excellent summary of the work.

Comment 1: Strengths: This simulation method produces accurate predictions when compared to experiments for the binding affinity of histones to DNA, counterion interactions, nucleosome DNA unwinding, nucleosome binding free energies, and sedimentation coefficients of arrays. The variety of measured quantities makes both this work and the impact of this coarse-grained methodology compelling. The comparison between the contributions of sodium and magnesium ions to nucleosome array compaction, presented in Figure 3, was exciting and a novel result that this simulation methodology can assess.

Response: We appreciate the reviewer’s strong assessment of the paper’s significance, novelty, and broad interest, and we thank him/her for the detailed suggestions and comments.

Comment 2: Weaknesses: The presentation of experimental data as representing in vivo systems is a simplification that may misrepresent the results of the simulation work. In vivo, in this context, typically means experimental data from whole cells. What one could expect for in vivo experimental data is measurements on nucleosomes from cell lysates where various and numerous chemical modifications are present. On the contrary, some of the experimental data used as a comparison are from in vitro studies. In vitro in this context means nucleosomes were formed ’in a test tube’ or under controlled conditions that do not represent the complexity of an in vivo system. The simulations performed here are more directly compared to in vitro conditions. This distinction likely impacts to what extent these simulation results are biologically relevant. In vivo and in vitro differences could be clarified throughout and discussed.

Response: As detailed in Response to Comment 3, we have made numerous modifications in the Introduction, Results, and Discussion Section to emphasize the differences between reconstituted and native nucleosomes. The newly added texts also delve into the utilization of the interaction strength measured for reconstituted nucleosomes as a reference point for conceptualizing the interactions among native nucleosomes.

Comment 3: In the introduction (pg. 3), the authors discuss the uncertainty of nucleosome-tonucleosome interaction strengths in vivo. For example, the authors discuss works such as Funke et al. However, Funke et al. used reconstituted nucleosomes from recombinant histones with one controlled modification (H4 acetylation). Therefore, this study that the authors discuss is measuring nucleosome’s in vitro affinity, and there could be significant differences in vivo due to various posttranslational modifications. Please revise the introduction, results section ”Close contacts drive nucleosome binding free energy,” and discussion to reflect and clarify the difference between in vitro and in vivo measurements. Please also discuss how biological variability could impact your findings in vivo. The works of Alexey Onufriev’s lab on the sensitivity of nucleosomes to charge changes (10.1016/j.bpj.2010.06.046, 10.1186/s13072-018-0181-5), such as some PTMs, are one potential starting place to consider how modifications alter nucleosome stability in vivo.

Response: We thank the reviewer for the insightful comments and agree that native nucleosomes can differ from reconstituted nucleosomes due to the presence of histone modifications.

We have revised the introduction to emphasize the differences between in vitro and in vivo nucleosomes. The new text now reads

"The relevance of physicochemical interactions between nucleosomes to chromatin organization in vivo has been constantly debated, partly due to the uncertainty in their strength [cite]. Examining the interactions between native nucleosomes poses challenges due to the intricate chemical modifications that histone proteins undergo within the nucleus and the variations in their underlying DNA sequences [cite]. Many in vitro experiments have opted for reconstituted nucleosomes that lack histone modifications and feature wellpositioned 601-sequence DNA to simplify the chemical complexity. These experiments aim to establish a fundamental reference point for understanding the strength of interactions within native nucleosomes. Nevertheless, even with reconstituted nucleosomes, a consensus regarding the significance of their interactions remains elusive. For example, using force-measuring magnetic tweezers, Kruithof et al. estimated the inter-nucleosome binding energy to be ∼ 14 kBT [cite]. On the other hand, Funke et al. introduced a DNA origamibased force spectrometer to directly probe the interaction between a pair of nucleosomes [cite], circumventing any potential complications from interpretations of single molecule traces of nucleosome arrays. Their measurement reported a much weaker binding free energy of approximately 2 kBT. This large discrepancy in the reported reference values complicates a further assessment of the interactions between native nucleosomes and their contribution to chromatin organization in vivo."

We modified the first paragraph of the results section to read

"Encouraged by the explicit ion model’s accuracy in reproducing experimental measurements of single nucleosomes and nucleosome arrays, we moved to directly quantify the strength of inter-nucleosomes interactions. We once again focus on reconstituted nucleosomes for a direct comparison with in vitro experiments. These experiments have yielded a wide range of values, ranging from 2 to 14 kBT [cite]. Accurate quantification will offer a reference value for conceptualizing the significance of physicochemical interactions among native nucleosomes in chromatin organization in vivo."

New text was added to the Discussion Section to emphasize the implications of simulation results for interactions among native nucleosomes.

"One significant finding from our study is the predicted strong inter-nucleosome interactions under the physiological salt environment, reaching approximately 9 kBT. We showed that the much lower value reported in a previous DNA origami experiment is due to the restricted nucleosomal orientation inherent to the device design. Unrestricted nucleosomes allow more close contacts to stabilize binding. A significant nucleosome binding free energy also agrees with the high forces found in single-molecule pulling experiments that are needed for chromatin unfolding [cite]. We also demonstrate that this strong inter-nucleosomal interaction is largely preserved at longer nucleosome repeat lengths (NRL) in the presence of linker histone proteins. While posttranslational modifications of histone proteins may influence inter-nucleosomal interactions, their effects are limited, as indicated by Ding et al. (Ding et al., 2021), and are unlikely to completely abolish the significant interactions reported here. Therefore, we anticipate that, in addition to molecular motors, chromatin regulators, and other molecules inside the nucleus, intrinsic inter-nucleosome interactions are important players in chromatin organization in vivo."

The suggested references (10.1016/j.bpj.2010.06.046, 10.1186/s13072-018-0181-5) are now included as citations # 44 and 45.

Comment 4: Due to the implicit water model, do you know if ions can penetrate the nucleosome more? For example, does the lack of explicit water potentially cause sodium to cluster in the DNA grooves more than is biologically relevant, as shown in Figure 1?

Response: We thank the reviewer for the insightful comments. The parameters of the explicit-ion model were deduced from all-atom simulations and fine-tuned to replicate crucial aspects of the local ion arrangements around DNA (1). The model’s efficacy was demonstrated in reproducing the radial distribution function of Na+ and Mg2+ ion distributions in the proximity of DNA (see Author response image 1). Consequently, the number of ions near DNA in the coarse-grained models aligns with that observed in all-atom simulations, and we do not anticipate any significant, unphysical clustering. It is worth noting that previous atomistic simulations have also reported the presence of a substantial quantity of Na+ ions in close proximity to nucleosomal DNA (refer to Author response image 2).

**Author response image 1. sa2fig1:** Comparison between the radial distribution functions of Na+ (left) and Mg2+ (right) ions around the DNA phosphate groups computed from all-atom (black) and coarse-grained (red) simulations. Figure reproduced from Figure 4 of [1]. The coarse-grained explicit ion model used in producing the red curves is identical to the one presented in the current manuscript.

**Author response image 2. sa2fig2:** Three-dimensional distribution of sodium ions around the nucleosome determined from all-atom explicit solvent simulations. Darker blue colors indicate higher sodium density and high density of sodium ions around the DNA is clearly visible. The crystallographically identified acidic patch has been highlighted as spheres on the surface of the histone core and a high level of sodium condensation is observed around these residues. Figure reproduced from [2].

Comment 5: Histone side chain to DNA interactions, such as histone arginines to DNA, are essential for nucleosome stability. Therefore, can the authors provide validation or references supporting your model of the nucleosome with one bead per amino acid? I would like to see if the nucleosomes are stable in an extended simulation or if similar dynamic motions to all-atom simulations are observed.

Response: The nucleosome model, which employs one bead per amino acid and lacks explicit ions, has undergone extensive calibration and has found application in numerous prior studies. For instance, the de Pablo group utilized a similar model to showcase its ability to accurately replicate the experimentally measured nucleosome unwinding free energy penalty (3), sequence-dependent nucleosome sliding (4), and the interaction between two nucleosomes (5). Similarly, the Takada group employed a comparable model to investigate acetylation-modulated tri-nucleosome structures (6), chromatin structures influenced by chromatin factors (7), and nucleosome sliding (8). Our group also employed this model to study the structural rearrangement of a tetranucleosome (9) and the folding of larger chromatin systems (10). In cases where data were available, simulations frequently achieved quantitative reproduction of experimental results.

We added the following text to the manuscript to emphasize previous studies that validate the model accuracy.

"We observe that residue-level coarse-grained models have been extensively utilized in prior studies to examine the free energy penalty associated with nucleosomal DNA unwinding [cite], sequence-dependent nucleosome sliding [cite], binding free energy between two nucleosomes [cite], chromatin folding [cite], the impact of histone modifications on tri-nucleosome structures [cite], and protein-chromatin interactions [cite]. The frequent quantitative agreement between simulation and experimental results supports the utility of such models in chromatin studies. Our introduction of explicit ions, as detailed below, further extends the applicability of these models to explore the dependence of chromatin conformations on salt concentrations."

We agree that arginines are important for nucleosome stability. Since we assign positive charges to these residues, their contribution to DNA binding can be effectively captured. The model’s ability in reproducing nucleosome stability is supported by the good agreement between the simulated free energy penalty associated with nucleosomal DNA unwinding and experimental value estimated from single molecule experiments (Figure 1).

To further evaluate nucleosome stability in our simulations, we conducted a 200-ns-long simulation of a nucleosome featuring the 601-sequence under physiological salt conditions– 100 mM NaCl and 0.5 mM MgCl2, consistent with the conditions in Figure 1 of the main text. We found that the nucleosome maintains its overall structure during this simulation. The nucleosome’s radius of gyration (Rg) remained proximate to the value corresponding to the PDB structure (3.95 nm) throughout the entire simulation period (see Author response image 3).

**Author response image 3. sa2fig3:** Time trace of the radius of gyration (Rg) of a nucleosome with the 601-sequence along an unbiased, equilibrium trajectory. It is evident the Rg fluctuates around the value found in the PDB structure (3.95 nm), supporting the stability of the nucleosome in our simulation.

Occasional fluctuations in Rg corresponded to momentary, partial unwrapping of the nucleosomal DNA, a phenomenon observed in single-molecule experiments. However, we advise caution due to the coarse-grained nature of our simulations, which prevents a direct mapping of simulation timescale to real time. Importantly, the rate of DNA unwrapping in our simulations is notably overestimated.

It’s plausible that coarse-grained models, lacking side chains, might underestimate the barrier for DNA sliding along the nucleosome. Specifically, our model, without differentiation between interactions among various amino acids and nucleotides, accurately reproduces the average nucleosomal DNA binding affinity but may not capture the energetic variations among binding interfaces. Since sliding’s contribution to chromatin organization is minimal due to the use of strongly positioning 601 sequences, we imposed rigidity on the two nucleotides situated at the dyad axis to prevent nucleosomal DNA sliding. In future studies, enhancing the calibration of protein-DNA interactions to achieve improved sequence specificity would be an intriguing avenue. To underscore this limitation of the model, we have included the following text in the discussion section of the main text.

"Several aspects of the coarse-grained model presented here can be further improved. For instance, the introduction of specific protein-DNA interactions could help address the differences in non-bonded interactions between amino acids and nucleotides beyond electrostatics [cite]. Such a modification would enhance the model’s accuracy in predicting interactions between chromatin and chromatin-proteins. Additionally, the single-bead-per-amino-acid representation used in this study encounters challenges when attempting to capture the influence of histone modifications, which are known to be prevalent in native nucleosomes. Multiscale simulation approaches may be necessary [cite]. One could first assess the impact of these modifications on the conformation of disordered histone tails using atomistic simulations. By incorporating these conformational changes into the coarse-grained model, systematic investigations of histone modifications on nucleosome interactions and chromatin organization can be conducted. Such a strategy may eventually enable the direct quantification of interactions among native nucleosomes and even the prediction of chromatin organization in vivo."

Comment 6: The solvent salt conditions vary in the experimental reference data for internucleosomal interaction energies. The authors note, for example, that the in vitro data from Funke et al. differs the most from other measurements, but the solvent conditions are 35 mM NaCl and 11 mM MgCl2. Since this simulation method allows for this investigation, could the authors speak to or investigate if solvent conditions are responsible for the variability in experimental reference data? The authors conclude on pg. 8-9 and Figure 4 that orientational restraints in the DNA origami methodology are responsible for differences in interaction energy. Can the authors rule out ion concentration contributions?

Response: We thank the reviewer for the insightful comment. We would like to clarify that the black curve presented in Figure 4B of the main text was computed using the salt concentration specified by Funke et al. (35 mM NaCl and 11 mM MgCl2). Furthermore, there were no restraints placed on nucleosome orientations during these calculations. Consequently, the results in Figure 4B can be directly compared with the black curve in Figure 5C. The data in Figure 5C were calculated under physiological salt conditions (150 mM NaCl and 2 mM MgCl2), which are the standard solvent salt conditions used in most studies.It is worth noting that the free energy of nucleosome binding is significantly higher at the salt concentration employed by Funke et al. (14 kBT) than the value at the physiological salt condition (9 kBT). Therefore, comparing the results in Figure 4B and 5C eliminates ion concentration conditions as a potential cause for the the almost negligible result reported by Funke et al.

Comment 7: In the discussion on pg. 12 residual-level should be residue-level.

Response: We apologize for the oversight and have corrected the grammatical error in our manuscript.

**RESPONSE TO REFEREE 2:**
Comment 0: In this manuscript, the authors introduced an explicit ion model using the coarse-grained modelling approach to model the interactions between nucleosomes and evaluate their effects on chromatin organization. The strength of this method lies in the explicit representation of counterions, especially divalent ions, which are notoriously difficult to model. To achieve their aims and validate the accuracy of the model, the authors conducted coarse-grained molecular dynamics simulations and compared predicted values to the experimental values of the binding energies of protein-DNA complexes and the free energy profile of nucleosomal DNA unwinding and inter-nucleosome binding. Additionally, the authors employed umbrella sampling simulations to further validate their model, reproducing experimentally measured sedimentation coefficients of chromatin under varying salt concentrations of monovalent and divalent ions.

Response: We thank the reviewer’s excellent summary of the work.

Comment 1: The significance of this study lies in the authors’ coarse-grained model which can efficiently capture the conformational sampling of molecules while maintaining a low computational cost. The model reproduces the scale and, in some cases, the shape of the experimental free energy profile for specific molecule interactions, particularly inter-nucleosome interactions. Additionally, the authors’ method resolves certain experimental discrepancies related to determining the strength of inter-nucleosomal interactions. Furthermore, the results from this study support the crucial role of intrinsic physicochemical interactions in governing chromatin organization within the nucleus.

Response: We appreciate the reviewer’s strong assessment of the paper’s significance, novelty, and broad interest, and we thank him/her for the detailed suggestions and comments.

Comment 2: The method is simple but can be useful, given the authors can provide more details on their ion parameterization. The paper says that parameters in their ”potentials were tuned to reproduce the radial distribution functions and the potential of mean force between ion pairs determined from all-atom simulations.” However, no details on their all-atom simulations were provided; at some point, the authors refer to Reference 67 which uses all-atom simulations but does not employ the divalent ions. Also, no explanation is given for their modelling of protein-DNA complexes.

Response: We appreciate the reviewer’s suggestion on clarifying the parameterization of the explicition model. The parameterization was not carried out in reference 67 nor by us, but by the de Pablo group in citation 53. Specifically, ion potentials were parameterized to fit the potential of mean force between both monovalent and divalent ion pairs, calculated either from all-atom simulations or from the literature. The authors carried out extensive validations of the model parameters by comparing the radial distribution functions of ions computed using the coarse-grained model with those from all-atom simulations. Good agreements between coarse-grained and all-atom results ensure that the parameters’ accuracy in reproducing the local structures of ion interactions.

To avoid confusion, we have revised the text from:

"Parameters in these potentials were tuned to reproduce the radial distribution functions and the potential of mean force between ion pairs determined from all-atom simulations."

to

"Parameters in these potentials were tuned by Freeman et al. [cite] to reproduce the radial distribution functions and the potential of mean force between ion pairs determined from all-atom simulations."

We modified the Supporting Information at several places to clarify the setup and interpretation of protein-DNA complex simulations.

For example, we clarified the force fields used in these simulation with the following text

"All simulations were carried out using the software Lammps [cite] with the force fields defined in the previous two sections."

We added details on the preparation of these simulations as follows

"We carried out a series of umbrella-sampling simulations to compute the binding free energies of a set of nine protein-DNA complexes with experimentally documented binding dissociation constants [cite]. Initial configurations of these simulations were prepared using the crystal structures with the corresponding PDB IDs listed in Fig. S1."

We further revised the caption of Figure S1 (included as Author response image 4) to facilitate the interpretation of simulation results.

**Author response image 4. sa2fig4:** The explicit-ion model predicts the binding affinities of protein-DNA complexes well, related to Fig. 1 of the main text. Experimental and simulated binding free energies are compared for nine protein-DNA complexes [cite], with a Pearson Correlation coefficient of 0.6. The PDB ID for each complex is indicated in red, and the diagonal line is drawn in blue. The significant correlation between simulated and experimental values supports the accuracy of the model. To further enhance the agreement between the two, it will be necessary to implement specific non-bonded interactions that can resolve differences among amino acids and nucleotides beyond simple electrostatics. Such modifications will be interesting avenues for future research. See text Section: Binding free energy of protein-DNA complexes for simulation details.

Comment 3: Overall, the paper is well-written, concise and easy to follow but some statements are rather blunt. For example, the linker histone contribution (Figure 5D) is not clear and could be potentially removed. The result on inter-nucleosomal interactions and comparison to experimental values from Ref#44 is the most compelling. It would be nice to see if the detailed shape of the profile for restrained inter-nucleosomal interactions in Figure 4B corresponds to the experimental profile. Including the dependence of free energy on a vertex angle would also be beneficial.

Response: We thank the reviewer for the comments and agree that the discussion on linker histone results was brief. However, we believe the results are important and demonstrate our model’s advantage over mesoscopic approaches in capturing the impact of chromatin regulators on chromatin organization.

Therefore, instead of removing the result, we expanded the text to better highlight its significance, to help its comprehension, and to emphasize its biological implications. The image in Figure 5D was also redesigned to better visualize the cross contacts between nucleosomes mediated by histone H1. The added texts are quoted as below, and the new Figure 5 is included.

**Author response image 5. sa2fig5:** Revised main text Figure 5, with Figure 5D modified for improved visual clarity.

"Importantly, we found that the weakened interactions upon extending linker DNA can be more than compensated for by the presence of histone H1 proteins. This is demonstrated in Fig. 5C and Fig. S8, where the free energy cost for tearing apart two nucleosomes with 167 bp DNA in the presence of linker histones (blue) is significantly higher than the curve for bare nucleosomes (red). Notably, at larger inter-nucleosome distances, the values even exceed those for 147 bp nucleosomes (black). A closer examination of the simulation configurations suggests that the disordered C-terminal tail of linker histones can extend and bind the DNA from the second nucleosome, thereby stabilizing the internucleosomal contacts (as shown in Fig. 5D). Our results are consistent with prior studies that underscore the importance of linker histones in chromatin compaction [cite], particularly in eukaryotic cells with longer linker DNA [cite]."

We further compared the simulated free energy profile, depicting the center of mass distance between nucleosomes, with the experimental profile, as depicted in Author response image 6. The agreement between the simulated and experimental results is evident. The nuanced features observed between 60 to 80 Å in the simulated profile stem from DNA unwinding to accommodate the incoming nucleosome, creating a small energy barrier. It’s worth noting that such unwinding is unlikely to occur in the experimental setup due to the hybridization method used to anchor nucleosomes onto the DNA origami. Moreover, our simulation did not encompass configurations below 60 Å, resulting in a lack of data in that region within the simulated profile.

We projected the free energy profile onto the vertex angle of the DNA origami device, utilizing the angle between two nucleosome faces as a proxy. Once more, the simulated profile demonstrates reasonable agreement with the experimental data (Author response image 6). Author response image 6 has been incorporated as Figure S4 in the Supporting Information.

**Author response image 6. sa2fig6:** Explicit ion modeling reproduces the experimental free energy profiles of nucleosome binding. (A) Comparison between the simulated (black) and experimental (red) free energy profile as a function of the inter-nucleosome distance. Error bars were computed as the standard deviation of three independent estimates. The barrier observed between 60Å and 80Å arises from the unwinding of nucleosomal DNA when the two nucleosomes are in close proximity, as highlighted in the orange circle. (B) Comparison between the simulated (black) and experimental (red) free energy profile as a function of the vertex angle. Error bars were computed as the standard deviation of three independent estimates. (C) Illustration of the vertex angle Φ used in panel (B).

Comment 4: Another limitation of this study is that the authors’ model sacrifices certain atomic details and thermodynamic properties of the modelled systems. The potential parameters of the counter ions were derived solely by reproducing the radial distribution functions (RDFs) and potential of mean force (PMF) based on all-atom simulations (see Methods), without considering other biophysical and thermodynamic properties from experiments. Lastly, the authors did not provide any examples or tutorials for other researchers to utilize their model, thus limiting its application.

Response: We agree that residue-level coarse-grained modeling indeed sacrifices certain atomistic details. This sacrifice can be potentially limiting when studying the impact of chemical modifications, especially on histone and DNA methylations. We added a new paragraph in the Discussion Section to point out such limitations and the relevant text is quoted below.

"Several aspects of the coarse-grained model presented here can be further improved. For instance, the introduction of specific protein-DNA interactions could help address the differences in non-bonded interactions between amino acids and nucleotides beyond electrostatics [cite]. Such a modification would enhance the model’s accuracy in predicting interactions between chromatin and chromatin-proteins. Additionally, the single-bead-per-amino-acid representation used in this study encounters challenges when attempting to capture the influence of histone modifications, which are known to be prevalent in native nucleosomes. Multiscale simulation approaches may be necessary [cite]. One could first assess the impact of these modifications on the conformation of disordered histone tails using atomistic simulations. By incorporating these conformational changes into the coarse-grained model, systematic investigations of histone modifications on nucleosome interactions and chromatin organization can be conducted. Such a strategy may eventually enable the direct quantification of interactions among native nucleosomes and even the prediction of chromatin organization in vivo."

Nevertheless, it’s important to note that while the model sacrifices accuracy, it compensates with superior efficiency. Atomistic simulations face significant challenges in conducting extensive free energy calculations required for a quantitative evaluation of ion impacts on chromatin structures.

The explicit ion model, introduced by the de Pablo group, follows a standard approach adopted by other research groups, such as the parameterization of ion models using the potential of mean force from atomistic simulations (11; 12). According to multiscale coarse-graining theory, reproducing potential mean force (PMF) enables the coarsegrained model to achieve thermodynamic consistency with the atomistic model, ensuring identical statistical properties derived from them. However, it’s crucial to recognize that an inherent limitation of such approaches is their dependence on the accuracy of atomistic force fields in reproducing thermodynamic properties from experiments, as any inaccuracies in the atomistic force fields will similarly affect the resulting coarse-grained (CG) model.

We have provided the implementation of CG model and detailed instructions on setting up and performing simulations GitHub repository. Examples include simulation setup for a protein-DNA complex and for a nucleosome with the 601-sequence.

References

[1] Freeman GS, Hinckley DM, de Pablo JJ (2011) A coarse-grain three-site-pernucleotide model for DNA with explicit ions. The Journal of Chemical Physics 135:165104.

[2] Materese CK, Savelyev A, Papoian GA (2009) Counterion Atmosphere and Hydration Patterns near a Nucleosome Core Particle. J. Am. Chem. Soc. 131:15005–15013.

[3] Lequieu J, Cordoba A, Schwartz DC, de Pablo JJ´ (2016) Tension-Dependent Free Energies of Nucleosome Unwrapping. ACS Cent. Sci. 2:660–666.

[4] Lequieu J, Schwartz DC, De Pablo JJ (2017) In silico evidence for sequence-dependent nucleosome sliding. Proc. Natl. Acad. Sci. U.S.A. 114.

[5] Moller J, Lequieu J, de Pablo JJ (2019) The Free Energy Landscape of Internucleosome Interactions and Its Relation to Chromatin Fiber Structure. ACS Cent. Sci. 5:341–348.

[6] Chang L, Takada S (2016) Histone acetylation dependent energy landscapes in trinucleosome revealed by residue-resolved molecular simulations. Sci Rep 6:34441.

[7] Watanabe S, Mishima Y, Shimizu M, Suetake I, Takada S (2018) Interactions of HP1 Bound to H3K9me3 Dinucleosome by Molecular Simulations and Biochemical Assays. Biophysical Journal 114:2336–2351.

[8] Brandani GB, Niina T, Tan C, Takada S (2018) DNA sliding in nucleosomes via twist defect propagation revealed by molecular simulations. Nucleic Acids Research 46:2788–2801.

[9] Ding X, Lin X, Zhang B (2021) Stability and folding pathways of tetra-nucleosome from six-dimensional free energy surface. Nat Commun 12:1091.

[10] Liu S, Lin X, Zhang B (2022) Chromatin fiber breaks into clutches under tension and crowding. Nucleic Acids Research 50:9738–9747.

[11] Savelyev A, Papoian GA (2010) Chemically accurate coarse graining of doublestranded DNA. Proc. Natl. Acad. Sci. U.S.A. 107:20340–20345.

[12] Noid WG (2013) Perspective: Coarse-grained models for biomolecular systems. The Journal of Chemical Physics 139:090901.